# COVERAGE AS A PRINCIPLE FOR DISCOVERING TRANSFERABLE BEHAVIOR IN REINFORCEMENT LEARNING

## ABSTRACT

Designing agents that acquire knowledge autonomously and use it to solve new tasks efficiently is an important challenge in reinforcement learning. Unsupervised learning provides a useful paradigm for autonomous acquisition of task-agnostic knowledge. In supervised settings, representations discovered through unsupervised pre-training offer important benefits when transferred to downstream tasks. Given the nature of the reinforcement learning problem, we explore how to transfer knowledge through behavior instead of representations. The behavior of pre-trained policies may be used for solving the task at hand (exploitation), as well as for collecting useful data to solve the problem (exploration). We argue that pre-training policies to maximize coverage will result in behavior that is useful for both strategies. When using these policies for both exploitation and exploration, our agents discover solutions that lead to larger returns. The largest gains are generally observed in domains requiring structured exploration, including settings where the behavior of the pre-trained policies is misaligned with the downstream task.

## 1 INTRODUCTION

Unsupervised representation learning techniques have led to unprecedented results in domains like computer vision (Hénaff et al., 2019; He et al., 2019) and natural language processing (Devlin et al., 2019; Radford et al., 2019). These methods are commonly composed of two stages – an initial unsupervised phase, followed by supervised fine-tuning on downstream tasks. The self-supervised nature of the learning objective allows to leverage large collections of unlabelled data in the first stage. This produces models that extract task-agnostic features that are well suited for transfer to downstream tasks. In reinforcement learning (RL), auxiliary representation learning objectives provide denser signals that result in data efficiency gains (Jaderberg et al., 2017) and even bridge the gap between learning from true state and pixel observations (Laskin et al., 2020). However, RL applications have not yet seen the advent of the two-stage setting where task-agnostic pre-training is followed by efficient transfer to downstream tasks. We argue that there are two reasons explaining this lag with respect to their supervised counterparts. First, these methods traditionally focus on transferring representations (Lesort et al., 2018). While this is enough in supervised scenarios, we argue that leveraging pre-trained *behavior* is far more important in RL domains requiring structured exploration. Second, what type of self-supervised objectives enable the acquisition of transferable, task-agnostic knowledge is still an open question. Defining these objectives in the RL setting is complex, as they should account for the fact that the the distribution of the input data will be defined by the behavior of the agent.

Transfer in deep learning is often performed through parameter initialization followed by fine-tuning. The most widespread procedure consists in initializing all weights in the neural network using those from the pre-trained model, and then adding an output layer with random parameters (Girshick et al., 2014; Devlin et al., 2019). Depending on the amount of available data, pre-trained parameters can either be fine-tuned or kept fixed. This builds on the intuition that the pre-trained model will map inputs to a feature space where the downstream task is easy to perform. In the RL setting, this procedure will completely dismiss the pre-trained policy and fall back to a random one when collecting experience. Given that complex RL problems require structured and temporally-extended behaviors, we argue that representation alone is not enough for efficient transfer in challenging

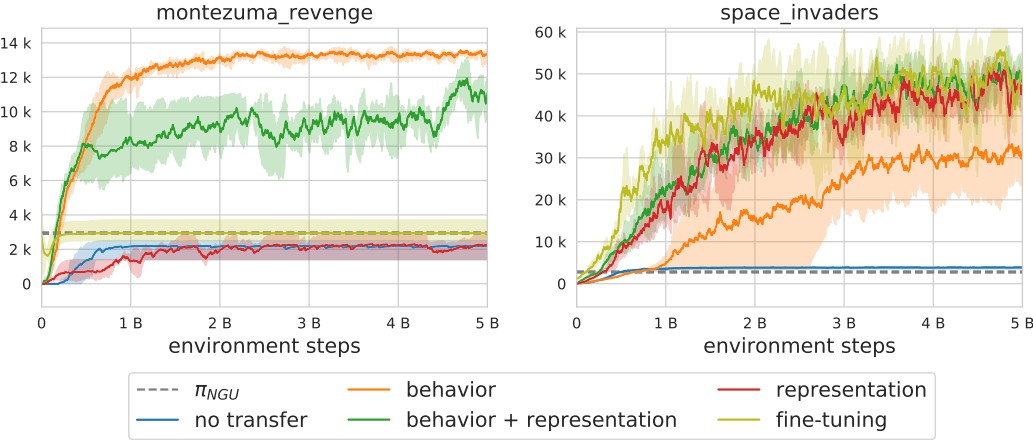

Figure 1: Comparison of transfer strategies on Montezuma's Revenge (hard exploration) and Space Invaders (dense reward) from a task-agnostic policy pre-trained with NGU (Puigdomènech Badia et al., 2020b). Transferring representations provides a significant boost on dense reward games, but it does not seem to help in hard exploration ones. Leveraging the behavior of the pre-trained policy provides important gains in hard exploration problems when compared to standard fine-tuning and is complementary to transferring representations. We refer the reader to Appendix F for details on the network architecture.

domains. Pre-trained representations do indeed provide data efficiency gains in domains with dense reward signals (Finn et al., 2017; Yarats et al., 2019; Stooke et al., 2020a), but our experiments show that the standard fine-tuning procedure falls short in hard exploration problems (c.f. Figure 1). We observe this limitation even when fine-tuning the pre-trained policy, which is aligned with findings from previous works (Finn et al., 2017). Learning in the downstream task can lead to catastrophically forgetting the pre-trained policy, something that depends on many difficult-to-measure factors such as the similarity between the tasks. We address the problem of leveraging arbitrary pre-trained policies when solving downstream tasks, a requirement towards enabling efficient transfer in RL.

Defining unsupervised RL objectives remains an open problem, and existing solutions are often influenced by how the acquired knowledge will be used for solving downstream tasks. Model-based approaches can learn world models from unsupervised interaction (Ha & Schmidhuber, 2018). However, the diversity of the training data will impact the accuracy of the model (Sekar et al., 2020) and deploying this type of approach in visually complex domains like Atari remains an open problem (Hafner et al., 2019). Unsupervised RL has also been explored through the lens of *empowerment* (Salge et al., 2014; Mohamed & Rezende, 2015), which studies agents that aim to discover intrinsic options (Gregor et al., 2016; Eysenbach et al., 2019). While these options can be leveraged by hierarchical agents (Florensa et al., 2017) or integrated within the universal successor features framework (Barreto et al., 2017; 2018; Borsa et al., 2019; Hansen et al., 2020), their lack of coverage generally limits their applicability to complex downstream tasks (Campos et al., 2020). We argue that maximizing coverage is a good objective for task-agnostic RL, as agents that succeed at this task will need to develop complex behaviors in order to efficiently explore the environment (Kearns & Singh, 2002). This problem can be formulated as that of finding policies that induce maximally entropic state distributions, which might become extremely inefficient in high-dimensional state spaces without proper priors (Hazan et al., 2019; Lee et al., 2019). In practice, exploration is often encouraged through intrinsic curiosity signals that incorporate priors in order to quantify how different the current state is from those already visited (Bellemare et al., 2016; Houthooft et al., 2016; Ostrovski et al., 2017; Puigdomènech Badia et al., 2020b). Agents that maximize these novelty-seeking signals have been shown to discover useful behaviors in unsupervised settings (Pathak et al., 2017; Burda et al., 2018a), but little research has been conducted towards leveraging the acquired knowledge once the agent is exposed to extrinsic reward. We show that coverage-seeking objectives are a good proxy for acquiring knowledge in task-agnostic settings, as leveraging the behaviors discovered in an unsupervised pre-training stage provides important gains when solving downstream tasks.

Our contributions can be summarized as follows. (1) We study how to transfer knowledge in RL through behavior by re-using pre-trained policies, an approach that is complementary to re-using representations. We argue that pre-trained behavior can be used for both exploitation and exploration, and present techniques to achieve both goals. (2) We propose coverage as a principle for discovering behavior that is suitable for both exploitation and exploration. While coverage is naturally aligned with exploration, we show that this objective will lead to the discovery of behavior that is useful for exploitation as well. (3) We propose *Coverage Pre-training for Transfer* (CPT), a method that implements the aforementioned hypotheses, and provide extensive experimental evaluation to support them. Our results show that leveraging the behavior of policies pre-trained to maximize coverage provides important benefits when solving downstream tasks. CPT obtains the largest gains in hard exploration games, where it almost doubles the median human normalized score achieved by our strongest baseline. Importantly, these benefits are observed even when the pre-trained policies are misaligned with the task being solved, confirming that the benefits do not come from a fortuitous alignment between our pre-training objective and the task reward. Furthermore, we show that CPT is able to leverage a single task-agnostic policy to solve multiple tasks in the same environment.

## 2 REINFORCEMENT LEARNING WITH UNSUPERVISED PRE-TRAINING

We follow a similar setup to that proposed by Hansen et al. (2020). In an initial pre-training stage, agents are allowed as many interactions with the environment as needed as long as they are not exposed to task-specific rewards. Rewards are reinstated in a second stage, where the knowledge acquired during unsupervised pre-training should be leveraged in order to enable efficient learning. This is analogous to the evaluation setting for unsupervised learning methods, where pre-training on classification benchmarks with labels removed is evaluated after fine-tuning on small sets of annotated examples.

The two-stage setup introduces two main challenges: defining pretext tasks in the absence of reward, and efficiently leveraging knowledge once rewards are reinstated. Our proposed method, *Coverage Pre-training for Transfer* (CPT), relies on coverage maximization as a pretext task for task-agnostic pre-training in order to produce policies whose behavior can be leveraged for both exploitation and exploration when solving downstream tasks in the same environment. Figure 2 provides intuition about the potential benefits of CPT.

Figure 2: Intuition behind CPT on a simple maze, where the agent needs to collect treasure chests (positive reward) while avoiding skulls (negative reward). Trajectories that a policy $\pi_p$ trained to maximize coverage could produce are depicted in orange. **Left:** while $\pi_p$ ignores some of the rewarding objects, many learning opportunities appear when following it during training. **Right:** combining primitive actions (red) with actions from $\pi_p$ (orange) side-steps the need to learn behavior that is already available through $\pi_p$ when solving downstream tasks.

## 3 LEVERAGING PRE-TRAINED POLICIES

Transfer in supervised domains often exploits the fact that related tasks might be solved using similar representations. This practice deals with the data inefficiency of training large neural networks with stochastic gradient descent. However, there is an additional source of data inefficiency when training RL agents: unstructured exploration. If the agent fails at discovering reward while exploring, it will struggle even when fitting simple function approximators on top of the true state of the MDP. These two strategies are complementary, as they address different sources of inefficiency, which motivates the study of techniques for leveraging pre-trained *behavior* (i.e. policies).

Our approach relies on off-policy learning methods in order to leverage arbitrary pre-trained policies. We make use of the mapping from observations to actions of such policies (i.e. their behavior), and do not transfer knowledge through pre-trained neural network weights. We consider value-based methods with experience replay that estimate action-value functions and derive greedy policies from them. The presented formulation considers a single pre-trained policy, $\pi_p$, but note that it is straightforward to extend it to multiple such policies. No assumptions are made on how the pre-trained policy is obtained, and it is only used for acting. We propose using the behavior of the pre-trained policy for two complementary purposes: *exploitation* and *exploration*. Figure 2 provides intuition about the potential benefits of these two approaches on a simple environment, and pseudo-code for the proposed methods is included in Appendix A.

**Exploitation.** When the behavior of $\pi_p$ is aligned with the downstream task, it can be used for zero-shot transfer. However, we are concerned with the more realistic scenario where only some of the behaviors of $\pi_p$ might be aligned with downstream tasks (c.f. Figure 2, right). We propose to leverage $\pi_p$ for exploitation by letting the agent combine primitive actions with the behavior of $\pi_p$. This is achieved by considering an expanded action set $\mathcal{A}^+ = \mathcal{A} \cup \{\pi_p(s)\}$, so that the agent can fall back to $\pi_p$ for one step when taking the additional action. Intuitively, this new state-dependent action should enable faster convergence when the pre-trained policy discovered behaviors that are useful for the task, while letting the agent ignore it otherwise. The return of taking action $a' \sim \pi_p(s)$ is used as target to fit both $Q(s, \pi_p(s))$ and $Q(s, a')$, which implements the observation that they are the same action and thus will lead to the same outcomes.

**Exploration.** Following the pre-trained policy might bring the agent to states that are unlikely to be visited with unstructured exploration techniques such as $\epsilon$-greedy. This property has the potential of accelerating learning even when the behavior of the pre-trained policy is not aligned with the downstream task, as it will effectively shorten the path between otherwise distant states (Liu & Brunskill, 2018). As we rely on off-policy methods that can learn from experience collected by arbitrary policies, we propose to perform temporally-extended exploration with $\pi_p$, which we will refer to as *flights*. Inspired by $\epsilon z$-greedy and its connection to Lévy flights (Viswanathan et al., 1996), a class of ecological models for animal foraging, these flights are started randomly and their duration is sampled from a heavy-tailed distribution. Our proposal can be understood as a variant of $\epsilon z$-greedy where pre-trained policies are used as exploration options. An exploratory flight might be started at any step with some probability. The duration for the flight is sampled from a heavy-tailed distribution, and control is handed over to $\pi_p$ during the complete flight. When not in a flight, the exploitative policy that maximizes the extrinsic reward is derived from the estimated Q-values using the $\epsilon$-greedy operator. This ensures that all state-action pairs will be visited given enough time, as exploring only with $\pi_p$ does not guarantee such property. Note that this is not needed in $\epsilon z$-greedy, which reduces to standard $\epsilon$-greedy exploration when sampling a flight duration of one step.

## 4    COVERAGE AS A GOAL FOR UNSUPERVISED PRE-TRAINING

So far we considered strategies for leveraging the behavior of arbitrary policies, and we now discuss how to train such policies in an initial pre-training stage with rewards removed. In such setting, it is a common practice to derive objectives for proxy tasks in order to drive learning. As we proposed to take advantage of pre-trained policies for both exploitation and exploration, it might seem unlikely that a single pre-training objective will produce policies that are useful for both purposes. However, we hypothesize that there exists a single criterion that will produce policies that can be used for both exploration and exploitation: *coverage*. This objective aims at visiting as many states as possible and is naturally aligned with exploration (Kearns & Singh, 2002). Long episodes where the agent visits as many different states as possible result in high returns in some domains such as videogames, locomotion and navigation (Pathak et al., 2017; Burda et al., 2018a). We argue that pre-training for coverage will bring benefits beyond these particular domains, as it fosters mastery over the environment. This leads to the discovery of skills and behaviors that can be exploited by the agent when solving downstream tasks even if the pre-trained policy does not obtain high returns.

Policies that maximize coverage should visit as many states as possible within a single episode, which differs from traditional exploration strategies employed when solving a single task. The goal of the latter is discovering potentially rewarding states, and the drive for exploration fades as strategies that lead to high returns are discovered. The proposed objective is closely related

to methods for task-agnostic exploration that train policies that induce maximally entropic state visitation distributions (Hazan et al., 2019; Lee et al., 2019). However, since the problems we are interested in involve large state spaces where states are rarely visited more than once, we instead propose to consider only the controllable aspects of the state space. This enables disentangling observations from states and gives rise to a more scalable, and thus more easily covered, notion of the state space.

We choose Never Give Up (NGU) (Puigdomènech Badia et al., 2020b) as a means for training policies that maximize coverage. NGU defines an intrinsic reward that combines per-episode and life-long novelty over controllable aspects of the state space. It can be derived directly from observations, unlike other approaches that make use of privileged information (Conti et al., 2018) or require estimating state visitation distributions (Hazan et al., 2019), making it suitable for environments that involve high-dimensional observations and partial observability. The intrinsic NGU reward maintains exploration throughout the entire training process, a property that makes it suitable for driving learning in task-agnostic settings. This contrasts with other intrinsic reward signals, that generally vanish as training progresses (Ecoffet et al., 2019). NGU was originally designed to solve hard-exploration problems by learning a family of policies with different degrees of exploratory behavior. Thanks to weight sharing, the knowledge discovered by exploratory policies enabled positive transfer to exploitative ones, obtaining impressive results when applied to large-scale domains (Puigdomènech Badia et al., 2020a). We instead propose to use NGU as a pre-training strategy in the absence of reward, transferring knowledge to downstream tasks in the form of behavior rather than weight sharing.

## 5   CPT: COVERAGE PRE-TRAINING FOR TRANSFER

CPT consists of two stages: (1) pre-training a task-agnostic policy using the intrinsic NGU reward, and (2) solving downstream tasks in the same environment by leveraging the pre-trained behavior.

**Coverage pre-training**

- The agent interacts with a Markov Decision Process (MDP) defined by the tuple $(\mathcal{S}, \mathcal{A}, P, r_{\text{NGU}}, \gamma)$, with $\mathcal{S}$ being the state space, $\mathcal{A}$ being the action space, $P$ the state-transition distribution, $\gamma \in (0, 1]$ the discount factor and the reward function $r_{NGU}$ is the intrinsic reward used in NGU (Puigdomènech Badia et al., 2020b).
- We use a value-based agent with a Q-function, $Q^{\text{NGU}}(s, a) : \mathcal{S} \times \mathcal{A} \to \mathbb{R}$, parameterised with a neural network as defined in Appendix F.
- We train $Q^{\text{NGU}}$ to maximise the NGU intrinsic reward, obtaining a deterministic policy given by $\pi_p(s) = \arg\max[Q^{\text{NGU}}(s, a)]$.
- We use $\epsilon$-greedy as behavioral policy when interacting with the environment.

**Transfer**

- We are given now a new MDP given by $(\mathcal{S}, \mathcal{A}, P, r, \gamma)$, where the only change with respect to the pre-training stage is a new extrinsic reward function $r : \mathcal{S} \times \mathcal{A} \to \mathbb{R}$.
- We define a Q-function $Q^{\pi}(s, a) : \mathcal{S} \times \mathcal{A}' \to \mathbb{R}$ on an extended action set $\mathcal{A}' = \mathcal{A} \cup a'$.
- When the agent selects action $a'$, it executes the action given by the pre-trained policy:

$$\pi(s) = \begin{cases} \arg\max_a[Q^{\pi}(s, a)] & \text{if } \arg\max_a[Q^{\pi}(s, a)] \neq a' \\ \pi_p(s) & \text{if } \arg\max_a[Q^{\pi}(s, a)] = a' \end{cases}$$

- We parameterise $Q^{\pi}$ using a neural network with random initialization.
- We use Lévy flights as behavioral policy when interacting with the environment. See Algorithm 3 for details.

## 6   EXPERIMENTS

We evaluate CPT in the Atari suite (Bellemare et al., 2013), a benchmark that presents a variety of challenges and is often used to measure the competence of agents. All our experiments are run using the distributed R2D2 agent (Kapturowski et al., 2019). A detailed description of the full distributed setting is provided in Appendix J. We use the same hyperparameters as in Agent57 (Puigdomènech Badia et al., 2020a), which are reported in Appendix B. All reported results are the average over three random seeds.

## 6.1 Unsupervised stage

Unsupervised RL methods are often evaluated by measuring the amount of task reward collected by the discovered policies (Burda et al., 2018a; Hansen et al., 2020), and we use this metric to evaluate the quality of our unsupervised policies. We pre-train our agents using 16B frames in order to guarantee the discovery of meaningful exploration policies[1], as it is common to let agents interact with the environment for as long as needed in this unsupervised stage (Hansen et al., 2020).

We compare the results of our unsupervised pre-training state against other unsupervised approaches, standard RL algorithms in the low-data regime and methods that perform unsupervised pre-training followed by an adaptation stage. We select some of the top performing methods in the literature, and refer the reader to Appendix C for a more extensive list of baselines. Since the NGU reward is non-negative, we consider a baseline where the agent obtains a constant positive reward at each step in order to measure the performance of policies that seek to stay alive for as long as possible. Table 1 shows that unsupervised CPT outperforms all baselines by a large margin, confirming the intuition that coverage is a good pre-training objective for the Atari benchmark. These results suggest that there is a strong correlation between exploration and the goals established by game designers (Burda et al., 2018a). In spite of the strong results, it is worth noting that unsupervised CPT achieves lower scores than random policies in some games, and it is quite inefficient at collecting rewards in some environments (e.g. it needs long episodes to obtain high scores). These observations motivate the development of techniques to leverage these pre-trained policies without compromising performance even when there exists a misalignment between objectives.

Table 1: Atari Suite comparisons. $@N$ represents the amount of RL interaction with reward utilized, with four frames observed at each iteration. *Mdn* and *M* are median and mean human normalized scores, respectively; $> 0$ is the number of games with better than random performance; and $> H$ is the number of games with human-level performance as defined in Mnih et al. (2015). **Top**: unsupervised learning only. **Mid**: data-limited RL. **Bottom**: RL with unsupervised pre-training.

| Algorithm | 26 Game Subset Kaiser et al. (2019) | | | | Full 57 Games Mnih et al. (2015) | | | |
|---|---|---|---|---|---|---|---|---|
| | Mdn | M | >0 | >H | Mdn | M | >0 | >H |
| Positive Reward R2D2 @0 | 9.44 | 59.55 | 21 | 4 | 3.46 | 45.23 | 46 | 5 |
| VISR @0 (Hansen et al., 2020) | 5.60 | 81.65 | 19 | 5 | 3.77 | 49.66 | 40 | 7 |
| CPT @0 | 80.92 | **494.54** | 25 | 12 | **81.72** | **320.06** | **52** | **27** |
| SimPLe @100$k$ (Kaiser et al., 2019) | 9.79 | 36.20 | 26 | 4 | – | – | – | – |
| DQN @200$M$ (Mnih et al., 2015) | **100.76** | 267.51 | **26** | **13** | 80.81 | 239.29 | 46 | 20 |
| GPI VISR @100$k$ (Hansen et al., 2020) | 6.59 | 111.23 | 22 | 7 | 8.99 | 109.16 | 44 | 12 |

## 6.2 Leveraging pre-trained policies

We now evaluate the proposed strategies for leveraging pre-trained policies once the reward function is reinstated by training R2D2-based agents (Kapturowski et al., 2019) for 5B frames. This is a relatively small budget for these distributed agents with hundreds of actors (Puigdomènech Badia et al., 2020a). We compare the proposed method against $\epsilon$-greedy and $\epsilon z$-greedy (Dabney et al., 2020) exploration strategies. Policies are evaluated using five parallel evaluator threads, and we report the average return over the last 300 evaluation episodes. Table 2 reports results in full Atari suite, which confirm the benefits of leveraging the behavior of a policy trained to maximize coverage. Our approach is most beneficial in the set of hard exploration games[2], where unstructured exploration generally precludes the discovery of high-performing policies.

It should be noted that our $\epsilon z$-greedy ablation under-performs relative to Dabney et al. (2020). This is due to our hyper-parameters and setting being derived from Puigdomènech Badia et al. (2020b), which adopts the standard Atari pre-processing (e.g. gray scale images and frame stacking). In

---

[1]The pre-training budget was not tuned, but we observe that competitive policies arise early in training. This observation suggests that smaller budgets are feasible as well.

[2]`montezuma_revenge`, `pitfall`, `private_eye`, `venture`, `gravitar`, `solaris`

contrast, Dabney et al. (2020) use color images, no frame stacking, a larger neural network and different hyper-parameters (e.g. smaller replay buffer). Studying if the performance of both NGU and the CPT is preserved in this setting is an important direction for future work. We suspect that improving the performance of our $\epsilon z$-greedy ablation will also improve our method, since exploration flights are central to both.

Table 2: Atari Suite comparisons for R2D2-based agents. $@N$ represents the amount of RL interaction with reward utilized, with four frames observed at each iteration. *Mdn*, *M* and *CM* are median, mean and mean capped human normalized scores, respectively.

| | Hard Exploration | | | Full 57 Games | | |
|---|---|---|---|---|---|---|
| Algorithm | Mdn | M | CM | Mdn | M | CM |
| $\epsilon$-greedy explore @$5B$ | 32.54 | 67.18 | 44.75 | 487.25 | 1753.81 | 90.32 |
| $\epsilon z$-greedy explore @$5B$ | 104.08 | 95.00 | 67.87 | 438.81 | 1263.83 | 92.53 |
| CPT (unsup) | 4.31 | 22.62 | 22.62 | 83.22 | 318.78 | 58.50 |
| CPT @$5B$ | **191.04** | **158.05** | **76.92** | **561.98** | **2184.26** | **93.20** |

**Ablation studies.** We run experiments on a subset of games in order to gain insight on the individual contribution of each of the proposed ways of leveraging the pre-trained policy. The subset is composed by 12 games[3], obtained by combining those used to tune hyperparameters by Hansen et al. (2020) with games where $\epsilon z$-greedy provides clear gains over $\epsilon$-greedy as per Dabney et al. (2020). This results in a set of games that require different amounts of exploration, and featuring both dense and sparse rewards. Figure 3 shows that both strategies obtain similar median scores across the 12 games, but combining them results in an important performance gain. This suggests that the gains they provide are complementary, and both are responsible for the strong performance of CPT. Note that CPT also outperforms a fine-tuning baseline, where the policy is initialized using the pre-trained weights rather than random ones. We believe that the benefits of both approaches can be combined by training via CPT a policy initialized with pre-trained weights.

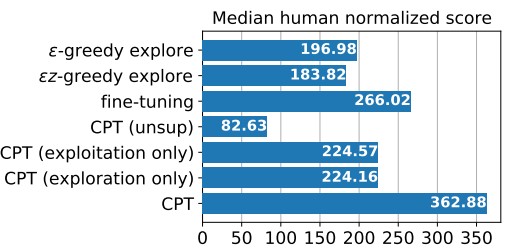

Figure 3: Ablation results. Using the task-agnostic policy for exploitation and exploration seems to provide complementary benefits, as combining the two techniques results in important gains.

**Effect of the pre-trained policy.** The behavior of the pre-trained policy will likely have a strong impact on the final performance of agents. We consider the amount of pre-training as a proxy for the exploration capabilities of the task-agnostic policies. Intuitively, policies trained for longer time spans will develop more complex behaviors that enable visiting a larger number of states. Figure 4 reports the end performance of agents after before and after transfer under different lengths of the pre-training phase, and shows how it has a different impact depending on the nature of the task. Montezuma's Revenge requires structured exploration for efficient learning, and longer pre-training times provide dramatic improvements in the end performance. Note that these improvements do not correlate with the task performance of the task-agnostic policy, which suggests that gains are due to a more efficient exploration of the state space. On the other hand, the final score in Pong is independent of the amount of pre-training. Simple exploration is enough to discover optimal policies, so the behaviors discovered by the unsupervised policy do not play an important role in this game.

**Transfer to multiple tasks.** An appealing property of task-agnostic knowledge is that it can be leveraged to solve multiple tasks. In the RL setting, this can be evaluated by leveraging a single task-agnostic policy for solving multiple tasks (i.e. reward functions) in the same environment. We evaluate whether the unsupervised NGU policies can be useful beyond the standard Atari tasks by creating two alternative versions of Ms Pacman and Hero with different levels of difficulty. The

---

[3] `asterix`, `bank_heist`, `frostbite`, `gravitar`, `jamesbond`, `montezuma_revenge`, `ms_pacman`, `pong`, `private_eye`, `space_invaders`, `tennis`, `up_n_down`.

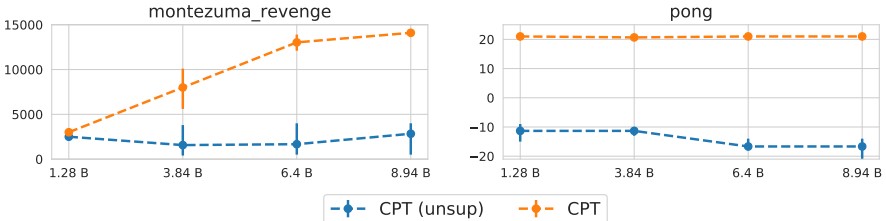

Figure 4: Effect of the pre-training budget, before and after adaptation, on Montezuma's Revenge (hard exploration) and Pong (dense reward).

goal in the modified version of Ms Pacman is to eat vulnerable ghosts, with pac-dots giving $0$ (easy version) or $-10$ (hard version) points. In the modified version of Hero, saving miners gives a fixed return of $1000$ points and dynamiting walls gives either $0$ (easy version) or $-300$ (hard version) points. The rest of rewards are removed, e.g. eating fruit in Ms Pacman or the bonus for unused power units in Hero. Note that even in the easy version of the games exploration is harder than in the original counterparts, as there are no small rewards guiding the agent towards its goals. In the hard version of the games exploration is even more challenging, as the intermediate rewards work as a deceptive signal that takes the agent away from its actual goal. In this case finding rewarding behaviors requires a stronger commitment to an exploration strategy. In this setting, the exploratory policies often achieve very low or even negative rewards, which contrasts with the strong performance they showed when evaluated under the standard game reward. Even in this adversarial scenario, results in Figure 5 shows that leveraging pre-trained exploration policies provides important gains. These results suggest that the strong performance observed under the standard game rewards is not due to an alignment between the NGU reward and the game goals, but due to an efficient usage of pre-trained exploration policies.

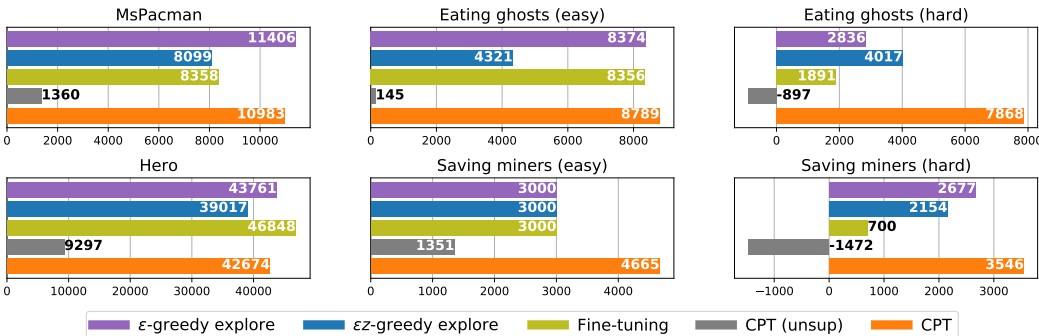

Figure 5: Final scores per task in the Atari games of Ms Pacman (top) and Hero (bottom) with modified reward functions. We train a single task-agnostic policy per environment, and leverage it to solve three different tasks: the standard game reward, a task with sparse rewards (easy), and a variant of the same task with deceptive rewards (hard). Despite the pre-trained policy might obtain low or even negative scores in some of the tasks, committing to its exploratory behavior eventually lets the agent discover strategies that lead to high returns.

**Towards the low-data regime.** So far we considered R2D2-based agents tuned for end-performance on massively distributed setups. Some applications might require higher efficiency in the low-data regime, even if this comes at the cost of a drop in end performance. The data efficiency of our method can be boosted by reusing representations from the pre-trained convolutional torso in the NGU policy (c.f. Figure 6 for details on the architecture), as shown in Figure 1. We observe that the data efficiency can be boosted further by decreasing the number of parallel actors. Figure 10 in the appendix showcases the improved data efficiency on Montezuma's Revenge when using 16 actors (instead of 256 as in previous experiments), obtaining superhuman scores in less than 50M frames. We note that this is around two times faster than the best results in the benchmark by Taïga et al. (2019), even though they consider single-threaded Rainbow-based agents (Hessel et al., 2018) that were designed for data efficiency.

## 7 RELATED WORK

Our work uses the experimental methodology presented in Hansen et al. (2020). But whereas that work only considered a simplified adaptation process that limited the final performance on the downstream task, the focus here is on the more general case of using a previously trained policy to aid in solving the full reinforcement learning problem. Specifically, VISR uses successor features to identify which of the pre-trained tasks best matches the true reward structure, which has previously been shown to work well for multi-task transfer (Barreto et al., 2018).

Gupta et al. (2018) provides an alternative method to meta-learn a solver for reinforcement learning problems from unsupervised reward functions. This method utilizes gradient-based meta-learning (Finn et al., 2017), which makes the adaptation process standard reinforcement learning updates. This means that even if the downstream reward is far outside of the training distribution, final performance would not necessarily be affected. However, these methods are hard to scale to the larger networks considered here, and followup work (Jabri et al., 2019) changed to memory-based meta-learning (Duan et al., 2016) which relies on information about rewards staying in the recurrent state. This makes it unsuitable to the sort of hard exploration problem our method excels at. Recent work has shown success in transferring representations learned in an unsupervised setting to reinforcement learning tasks (Stooke et al., 2020b). Our representation transfer experiments suggest that this should handicap final performance, but the possibility also exists that different unsupervised objectives should be used for representation transfer and policy transfer.

Concurrent work by Bagot et al. (2020) also augments an agent with the ability to utilize another policy. However, their work treats the unsupervised policy as an option, only callable for an extended duration. In contrast, we only perform extended calls to the unsupervised policy during exploratory levy flights and augment the action space to allow for single time-step calls. This difference between exploratory and exploitative calls to the unsupervised policy in critical to overall performance, as illustrated in Figure 3. In addition, in Bagot et al. (2020) the unsupervised policy is learned in tandem based on an intrinsic reward function. This is a promising direction which is complementary to our work, as it handles the case wherein there is no unsupervised pre-training phase. However, their work only considers tabular domains, so it is unclear how this approach would fair in the high-dimensional state spaces considered here.

## 8 DISCUSSION

We studied the problem of transferring pre-trained behavior in reinforcement learning, an approach that is complementary to the common practice of transferring representations. Depending on the behavior of the pre-trained policies, we argued that they might be useful for exploitation, exploration, or both. We proposed methods to make use of pre-trained behavior for both purposes: exploiting with the pre-trained policy by making it available to the agent as an extra action, and performing temporally-extended exploration with it. While we make no assumption on the nature of the pre-trained policies, this raises the question of how to discover behaviors that are suitable for transfer. We proposed coverage as a principle for pre-training task-agnostic policies that are suitable for both exploitation and exploration. We chose NGU in our experiments for its scalability, but note that our approach could be combined with any other strategy for maximizing coverage. We found that unsupervised training with this objective produces strong performing policies in the Atari suite, likely due to the way in which the goals in some of these tasks were designed (Burda et al., 2018a). Our transfer experiments demonstrate that these pre-trained policies can be used to boost the performance of agents trained to maximize reward, providing the most important gains in hard exploration tasks. These benefits are not due to an alignment between our pre-training and downstream tasks, as we also observed positive transfer in games where the pre-trained policy obtained low scores. In order to provide further evidence for this claim, we designed alternative tasks for Atari games involving hard exploration and deceptive rewards. Our transfer strategy outperformed all considered baselines in these settings, even when the pre-trained policy obtained very low or even negative scores, demonstrating the generality of the method. Besides disambiguating the role of the alignment between pre-training and downstream tasks, these experiments demonstrate the utility of a single task-agnostic policy for solving multiple tasks in the same environment.

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

# A   PSEUDO-CODE

Algorithm 1 provides pseudo-code for the flight logic that controls how the pre-trained policy is used for exploration purposes. At each step, a flight is started with probability $\epsilon_{\text{levy}}$. The duration for the flight is sampled from a heavy-tailed distribution, $n_{\text{dist}}$, similarly to $\epsilon z$-greedy (c.f. Appendix B for more details). When not in a flight, the exploitative policy that maximizes the extrinsic reward is derived from the estimated Q-values using the $\epsilon$-greedy operator. This ensures that all state-action pairs will be visited given enough time, as exploring only with $\pi_p$ does not guarantee such property. Note that this is not needed in $\epsilon z$-greedy, which reduces to standard $\epsilon$-greedy exploration when the sampled flight duration equals 1.

Algorithm 2 provides pseudo-code for the actor logic when using the augmented action set, $\mathcal{A}^+ = \mathcal{A} \cup \{\pi_p(s)\}$. It derives an $\epsilon$-greedy policy over $|\mathcal{A}| + 1$ actions, where the $(|\mathcal{A}| + 1)$-th action is resolved by sampling from $\pi_p(s)$.

Finally, Algorithm 3 provides pseudo-code for the actor in the full CPT method that combines Algorithms 1 and 2.

---

**Algorithm 1:** Actor pseudo-code for CPT (exploration only)

---

**Input:** Q-value estimate for the current policy, $Q^\pi(s,a)$
**Input:** Pre-trained policy, $\pi_p$
**Input:** Probability of starting a flight, $\epsilon_{\text{levy}}$
**Input:** Flight length distribution, $n_{\text{dist}}$
**while** *True* **do**

    $n \leftarrow 0$                       `// flight length`
    **while** episode not ended **do**

        Observe state $s$
        **if** $n == 0$ and *random()* $\leq \epsilon_{levy}$ **then**
            $n \sim n_{\text{dist}}$
        **end**
        **if** $n > 0$ **then**
            $n \leftarrow n - 1$
            $a \leftarrow \pi_p(s)$              `// explore with` $\pi_p$
        **else**
            $a \leftarrow \epsilon\text{-greedy}[Q^\pi(s,a)]$
        **end**
        Take action $a$
    **end**

**end**

---

**Algorithm 2:** Actor pseudo-code for CPT (exploitation only)

---

**Input:** Action set $\mathcal{A}$
**Input:** Pre-trained policy, $\pi_p$
**Input:** Q-value estimate for the current policy, $Q^\pi(s,a) \; \forall a \in \mathcal{A} \cup \{\pi_p(s)\}$
**Input:** Probability of taking an exploratory action, $\epsilon$
**while** *True* **do**

    **while** episode not ended **do**

        Observe state $s$
        **if** *random()* $\leq \epsilon$ **then**
            $a \leftarrow \text{Uniform}(1, |\mathcal{A}| + 1)$
        **else**
            $a \leftarrow \arg\max[Q^\pi(s,a)]$
        **end**
        **if** $a == |\mathcal{A}| + 1$ **then**
            $a \leftarrow \pi_p(s)$              `// exploit with` $\pi_p$
        **end**
        Take action $a$
    **end**

**end**

---

**Algorithm 3:** Actor pseudo-code for CPT

**Input:** Action set $\mathcal{A}$
**Input:** Pre-trained policy, $\pi_p$
**Input:** Q-value estimate for the current policy, $Q^\pi(s, a) \, \forall a \in \mathcal{A} \cup \{\pi_p(s)\}$
**Input:** Probability of taking an exploratory action, $\epsilon$
**Input:** Probability of starting a flight, $\epsilon_{\text{levy}}$
**Input:** Flight length distribution, $n_{\text{dist}}$
**while** *True* **do**
    $n \leftarrow 0$                      `// flight length`
    **while** episode not ended **do**
        Observe state $s$
        **if** $n == 0$ and *random()* $\leq \epsilon_{levy}$ **then**
            $n \sim n_{\text{dist}}$
        **end**
        **if** $n > 0$ **then**
            $n \leftarrow n - 1$
            $a \leftarrow \pi_p(s)$           `// explore with `$\pi_p$
        **else**
            **if** *random()* $\leq \epsilon$ **then**
                $a \leftarrow \text{Uniform}(1, |\mathcal{A}| + 1)$
            **else**
                $a \leftarrow \arg\max[Q^\pi(s, a)]$
            **end**
            **if** $a == |\mathcal{A}| + 1$ **then**
                $a \leftarrow \pi_p(s)$         `// exploit with `$\pi_p$
            **end**
        **end**
        Take action $a$
    **end**
**end**

## B  HYPERPARAMETERS

Table 3 summarizes the main hyperparameters of our method. The pre-trained policies were optimized using Retrace (Munos et al., 2016). Transfer was performed with Peng's $Q(\lambda)$ (Peng & Williams, 1994) instead, which we found to be much more data efficient in our experiments. The reason for this difference is that the benefits of $Q(\lambda)$ were observed once unsupervised policies had been trained on all Atari games, and we suspect that the data efficiency gains will transfer to the pre-training stage as well.

Table 3: Hyperparameter values used in R2D2-based agents. The rest of hyperparameters use the values reported by Kapturowski et al. (2019).

| Hyperparameter | Value |
|---|---|
| Number of actors | 256 |
| Actor parameter update interval | 400 environment steps |
| Sequence length | 160 (without burn-in) |
| Replay buffer size | $12.5 \times 10^4$ part-overlapping sequences |
| Priority exponent | 0.9 |
| Importance sampling exponent | 0 |
| Learning rule (downstream tasks) | $Q(\lambda)$, $\lambda = 0.7$ |
| Learning rule (NGU pre-training) | Retrace$(\lambda)$, $\lambda = 0.95$ |
| Discount (downstream tasks) | 0.99 |
| Discount (NGU pre-training) | 0.97 |
| Minibatch size | 64 |
| Optimizer | Adam |
| Optimizer settings | $\varepsilon = 10^{-4}$, $\beta_1 = 0.9$, $\beta_2 = 0.999$ |
| Learning rate | $2 \times 10^{-4}$ |
| Target network update interval | 1500 updates |
| $\epsilon_{\text{levy}}$ distribution | Log-Uniform$[0, 0.1]$ |
| Flight length distribution | Zeta with $\mu = 2$ |

## C   EXTENDED UNSUPERVISED RL RESULTS

Table 4 compares unsupervised CPT with all the methods reported by Hansen et al. (2020).

Table 4: Atari Suite comparisons, adapted from Hansen et al. (2020). $@N$ represents the amount of RL interaction with reward utilized, with four frames observed at each iteration. *Mdn* and *M* are median and mean human normalized scores, respectively; $> 0$ is the number of games with better than random performance; and $> H$ is the number of games with human-level performance as defined in Mnih et al. (2015). **Top**: unsupervised learning only. **Mid**: data-limited RL. **Bottom**: RL with unsupervised pre-training.

| | 26 Game Subset Kaiser et al. (2019) | | | | 47 Game Subset Burda et al. (2018a) | | | | Full 57 Games Mnih et al. (2015) | | | |
|---|---|---|---|---|---|---|---|---|---|---|---|---|
| Algorithm | Mdn | M | >0 | >H | Mdn | M | >0 | >H | Mdn | M | >0 | >H |
| IDF Curiosity @0 | – | – | – | – | 8.46 | 24.51 | 34 | 5 | – | – | – | – |
| RF Curiosity @0 | – | – | – | – | 7.32 | 29.03 | 36 | 6 | – | – | – | – |
| Pos Reward NSQ @0 | 2.18 | 50.33 | 14 | 5 | 0.69 | 57.65 | 26 | 8 | 0.29 | 41.19 | 28 | 8 |
| Pos Reward R2D2 @0 | 9.44 | 59.55 | 21 | 4 | 14.16 | 57.53 | 39 | 5 | 3.46 | 45.23 | 46 | 5 |
| Q-DIAYN-5 @0 | 0.17 | −3.60 | 13 | 0 | 0.33 | −1.23 | 25 | 2 | 0.34 | −2.18 | 30 | 2 |
| Q-DIAYN-50 @0 | −1.65 | −21.77 | 4 | 0 | −1.69 | −16.26 | 8 | 0 | −3.16 | −20.31 | 9 | 0 |
| VISR @0 | 5.60 | 81.65 | 19 | 5 | 4.04 | 58.47 | 35 | 7 | 3.77 | 49.66 | 40 | 7 |
| CPT @0 | 80.92 | **494.54** | 25 | 12 | **96.10** | **310.27** | **45** | **23** | **81.72** | **320.06** | **52** | **27** |
| SimPLe @100$k$ | 9.79 | 36.20 | 26 | 4 | – | – | – | – | – | – | – | – |
| DQN @10$M$ | 27.80 | 52.95 | 25 | 7 | 9.91 | 28.07 | 41 | 7 | 8.61 | 27.55 | 48 | 7 |
| DQN @200$M$ | **100.76** | 267.51 | **26** | **13** | – | – | – | – | 80.81 | 239.29 | 46 | 20 |
| Rainbow @100$k$ | 2.23 | 10.12 | 25 | 1 | – | – | – | – | – | – | – | – |
| PPO @500$k$ | 20.93 | 43.74 | 25 | 7 | – | – | – | – | – | – | – | – |
| NSQ @10$M$ | 8.20 | 33.80 | 22 | 3 | 7.29 | 29.47 | 37 | 4 | 6.80 | 28.51 | 43 | 5 |
| Q-DIAYN-5 @100$k$ | 0.01 | 16.94 | 13 | 2 | 1.31 | 19.64 | 28 | 6 | 1.55 | 16.65 | 33 | 6 |
| Q-DIAYN-50 @100$k$ | −1.64 | −27.88 | 3 | 0 | −1.66 | −16.74 | 8 | 0 | −2.53 | −24.13 | 9 | 0 |
| RF VISR @100$k$ | 7.24 | 58.23 | 20 | 6 | 3.81 | 42.60 | 33 | 9 | 2.16 | 35.29 | 39 | 9 |
| VISR @100$k$ | 9.50 | 128.07 | 21 | 7 | 9.42 | 121.08 | 35 | 11 | 6.81 | 102.31 | 40 | 11 |
| GPI RF VISR @100$k$ | 5.55 | 58.77 | 20 | 5 | 4.24 | 48.38 | 34 | 9 | 3.60 | 40.01 | 40 | 10 |
| GPI VISR @100$k$ | 6.59 | 111.23 | 22 | 7 | 11.70 | 129.76 | 38 | 12 | 8.99 | 109.16 | 44 | 12 |

## D   EXTENDED ATARI57 RESULTS

Table 5: Atari Suite comparisons for R2D2-based agents. $@N$ represents the amount of RL interaction with reward utilized, with four frames observed at each iteration. *Mdn*, *M* and *CM* are median, mean and mean capped human normalized scores, respectively.

| | Hard Exploration | | | Full 57 Games | | |
|---|---|---|---|---|---|---|
| Algorithm | Mdn | M | CM | Mdn | M | CM |
| $\epsilon$-greedy explore @1$B$ | 31.07 | 39.40 | 34.75 | 229.75 | 864.69 | 84.56 |
| $\epsilon z$-greedy explore @1$B$ | 42.55 | 53.90 | 46.21 | 204.52 | 578.73 | 85.11 |
| CPT @1$B$ | **100.89** | **94.20** | **63.95** | **273.49** | **1517.13** | **86.38** |
| $\epsilon$-greedy explore @5$B$ | 32.54 | 67.18 | 44.75 | 487.25 | 1753.81 | 90.32 |
| $\epsilon z$-greedy explore @5$B$ | 104.08 | 95.00 | 67.87 | 438.81 | 1263.83 | 92.53 |
| CPT @5$B$ | **191.04** | **158.05** | **76.92** | **561.98** | **2184.26** | **93.20** |

# E ALTERNATIVE REWARD FUNCTIONS

**MsPacman: eating ghosts**

- Pac-dots: 0 points (easy) or -10 points (hard)
- Eating vulnerable ghosts:
    - #1 in succession: 200 points
    - #2 in succession: 400 points
    - #3 in succession: 800 points
    - #4 in succession: 1600 points
- Other actions: 0 points

**Hero: rescuing miners**

- Dynamiting walls: 0 points (easy) or -300 points (hard)
- Rescuing a miner: 1000 points
- Other actions: 0 points

# F Q-NETWORK ARCHITECTURE

All policies use the same Q-Network architecture as Agent57 (Puigdomènech Badia et al., 2020a), which is composed by a convolutional torso followed by an LSTM (Hochreiter & Schmidhuber, 1997) and a dueling head (Wang et al., 2016). When leveraging the behavior of the pre-trained policy to solve new tasks, one can train a new policy from scratch or share some of the components for increased efficiency (c.f. Figure 6). Shared weights are kept fixed in order to preserve the behavior of the pre-trained policy.

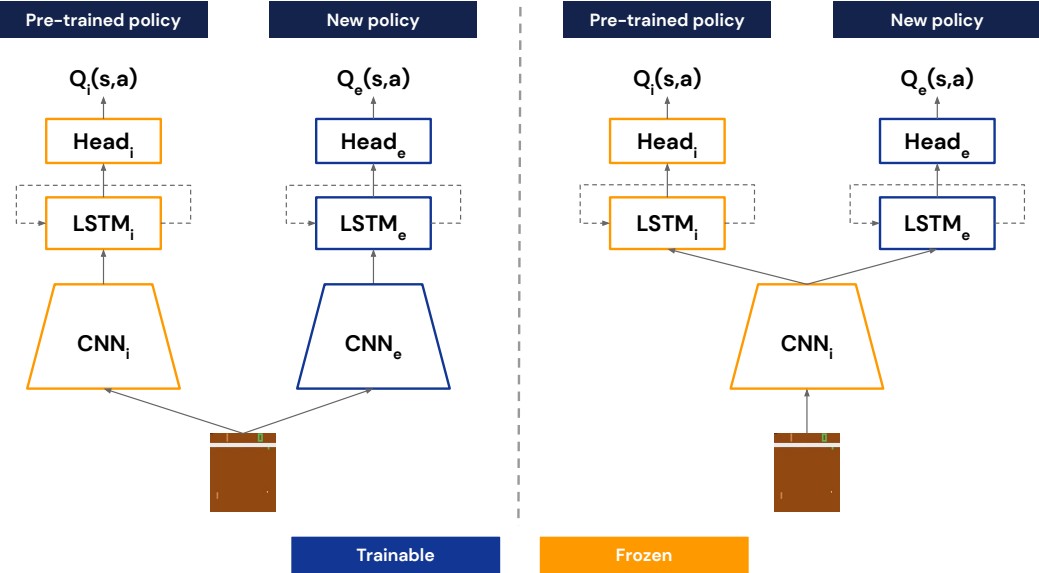

Figure 6: Q-Network architecture for the reinforcement learning stage. The pre-trained policy can be leveraged without transferring representations (left), but sharing weights generally provides efficiency gains early in training (right).

# G  SCORES PER GAME

Table 6: Results per game at 5B training frames.

| Game | $\epsilon$-greedy explore | $\epsilon z$-greedy explore | CPT |
|------|---------------------------|------------------------------|-----|
| alien | $10831.17 \pm 2114.29$ | $14634.02 \pm 1109.15$ | $\mathbf{15657.57 \pm 1717.96}$ |
| amidar | $\mathbf{11761.67 \pm 1560.86}$ | $6784.28 \pm 718.05$ | $10394.96 \pm 891.60$ |
| assault | $\mathbf{15940.72 \pm 3531.69}$ | $9177.28 \pm 2170.26$ | $15060.31 \pm 740.63$ |
| asterix | $472812.21 \pm 222663.81$ | $374966.62 \pm 135810.51$ | $\mathbf{630663.91 \pm 82753.46}$ |
| asteroids | $45716.28 \pm 3642.38$ | $\mathbf{147005.85 \pm 44313.45}$ | $31957.42 \pm 15540.09$ |
| atlantis | $\mathbf{1514724.43 \pm 10941.36}$ | $1132188.04 \pm 43551.36$ | $1491384.23 \pm 5978.05$ |
| bank heist | $965.63 \pm 133.72$ | $1058.75 \pm 135.46$ | $\mathbf{13913.32 \pm 3529.15}$ |
| battle zone | $292553.41 \pm 18196.77$ | $\mathbf{312367.76 \pm 43554.18}$ | $258533.57 \pm 22865.64$ |
| beam rider | $18472.45 \pm 1977.78$ | $\mathbf{22403.95 \pm 1596.92}$ | $16301.02 \pm 1853.73$ |
| berzerk | $\mathbf{12343.83 \pm 3331.54}$ | $3846.56 \pm 1723.24$ | $8359.80 \pm 201.10$ |
| bowling | $141.64 \pm 4.52$ | $156.32 \pm 8.11$ | $\mathbf{174.27 \pm 0.10}$ |
| boxing | $99.96 \pm 0.03$ | $99.94 \pm 0.06$ | $\mathbf{100.00 \pm 0.00}$ |
| breakout | $432.65 \pm 27.35$ | $393.19 \pm 35.12$ | $\mathbf{441.21 \pm 15.08}$ |
| centipede | $189502.66 \pm 31388.08$ | $\mathbf{358841.20 \pm 73578.20}$ | $178635.17 \pm 17227.15$ |
| chopper command | $611393.11 \pm 65206.69$ | $\mathbf{697655.53 \pm 215090.74}$ | $573055.88 \pm 75343.57$ |
| crazy climber | $\mathbf{229992.57 \pm 17738.33}$ | $212001.76 \pm 1853.07$ | $226821.26 \pm 3608.19$ |
| defender | $\mathbf{547238.15 \pm 2579.38}$ | $516521.06 \pm 11969.59$ | $540124.74 \pm 4488.40$ |
| demon attack | $143662.42 \pm 88.16$ | $141352.18 \pm 3848.73$ | $\mathbf{143762.91 \pm 106.75}$ |
| double dunk | $\mathbf{23.99 \pm 0.02}$ | $23.88 \pm 0.06$ | $23.85 \pm 0.15$ |
| enduro | $2358.37 \pm 3.32$ | $2359.08 \pm 1.03$ | $\mathbf{2361.56 \pm 1.03}$ |
| fishing derby | $12.80 \pm 77.79$ | $\mathbf{64.74 \pm 0.59}$ | $52.58 \pm 0.32$ |
| freeway | $\mathbf{33.87 \pm 0.08}$ | $33.77 \pm 0.03$ | $33.79 \pm 0.08$ |
| frostbite | $9287.24 \pm 167.11$ | $8504.41 \pm 940.72$ | $\mathbf{17692.42 \pm 2871.83}$ |
| gopher | $\mathbf{117398.58 \pm 2485.82}$ | $84140.40 \pm 12919.83$ | $113716.78 \pm 3966.91$ |
| gravitar | $6123.08 \pm 103.19$ | $5798.68 \pm 735.59$ | $\mathbf{8373.70 \pm 1260.75}$ |
| hero | $\mathbf{46048.07 \pm 6970.26}$ | $39700.22 \pm 4379.84$ | $40825.09 \pm 3736.25$ |
| ice hockey | $32.43 \pm 30.64$ | $30.65 \pm 28.17$ | $\mathbf{60.36 \pm 4.94}$ |
| jamesbond | $\mathbf{6056.14 \pm 1643.52}$ | $3843.92 \pm 118.35$ | $1484.87 \pm 489.66$ |
| kangaroo | $14672.37 \pm 187.16$ | $14730.99 \pm 114.20$ | $\mathbf{15965.79 \pm 36.61}$ |
| krull | $10081.04 \pm 594.10$ | $10171.52 \pm 399.81$ | $\mathbf{406596.00 \pm 55547.76}$ |
| kung fu master | $\mathbf{200721.64 \pm 2265.35}$ | $171591.29 \pm 8516.87$ | $196638.89 \pm 456.09$ |
| montezuma revenge | $1478.38 \pm 1114.20$ | $1467.77 \pm 1104.72$ | $\mathbf{12086.71 \pm 1217.76}$ |
| ms pacman | $\mathbf{11212.85 \pm 103.23}$ | $7511.39 \pm 406.77$ | $10996.90 \pm 262.74$ |
| name this game | $32138.12 \pm 2156.95$ | $\mathbf{37343.04 \pm 1917.73}$ | $30252.11 \pm 884.84$ |
| phoenix | $\mathbf{712101.72 \pm 62738.09}$ | $80611.18 \pm 25316.56$ | $553429.34 \pm 24278.55$ |
| pitfall | $\mathbf{-0.19 \pm 0.15}$ | $-12.34 \pm 4.20$ | $-0.39 \pm 0.39$ |
| pong | $\mathbf{20.93 \pm 0.01}$ | $20.49 \pm 0.10$ | $20.90 \pm 0.01$ |
| private eye | $23592.22 \pm 11876.55$ | $\mathbf{50770.82 \pm 14984.92}$ | $40435.54 \pm 51.04$ |
| qbert | $\mathbf{24343.75 \pm 1904.89}$ | $16975.13 \pm 1332.44$ | $16057.31 \pm 318.87$ |
| riverraid | $\mathbf{32325.07 \pm 1185.15}$ | $30582.53 \pm 638.47$ | $28550.32 \pm 2298.03$ |
| road runner | $\mathbf{423191.07 \pm 53071.15}$ | $88890.04 \pm 24971.18$ | $251261.09 \pm 31741.38$ |
| robotank | $97.23 \pm 1.22$ | $\mathbf{108.92 \pm 4.79}$ | $98.45 \pm 2.85$ |
| seaquest | $\mathbf{188771.84 \pm 20759.57}$ | $175745.09 \pm 120718.82$ | $86605.86 \pm 55065.85$ |
| skiing | $\mathbf{-29854.11 \pm 85.79}$ | $-30060.81 \pm 142.32$ | $-30121.95 \pm 70.62$ |
| solaris | $17741.02 \pm 5340.46$ | $16127.73 \pm 2975.20$ | $\mathbf{24366.59 \pm 4868.05}$ |
| space invaders | $3621.76 \pm 5.81$ | $3547.78 \pm 35.31$ | $\mathbf{30609.21 \pm 7141.11}$ |
| star gunner | $\mathbf{223536.63 \pm 48548.34}$ | $179698.69 \pm 12194.36$ | $171294.31 \pm 23185.79$ |
| surround | $\mathbf{8.24 \pm 0.48}$ | $1.48 \pm 8.12$ | $5.86 \pm 1.44$ |
| tennis | $7.99 \pm 22.56$ | $7.98 \pm 22.51$ | $\mathbf{23.96 \pm 0.01}$ |
| time pilot | $\mathbf{139931.67 \pm 70521.78}$ | $71768.84 \pm 2933.22$ | $44936.87 \pm 137.49$ |
| tutankham | $324.02 \pm 4.26$ | $311.65 \pm 8.62$ | $\mathbf{420.36 \pm 30.13}$ |
| up n down | $529363.05 \pm 16813.20$ | $394984.70 \pm 34313.42$ | $\mathbf{562739.02 \pm 8527.59}$ |
| venture | $0.00 \pm 0.00$ | $1833.85 \pm 43.73$ | $\mathbf{2110.64 \pm 55.39}$ |
| video pinball | $454023.46 \pm 377076.03$ | $107071.98 \pm 67142.18$ | $\mathbf{463141.28 \pm 426927.92}$ |
| wizard of wor | $\mathbf{40833.65 \pm 4776.81}$ | $38275.31 \pm 4177.41$ | $30453.12 \pm 2470.20$ |
| yars revenge | $279765.86 \pm 27370.20$ | $250483.70 \pm 54593.32$ | $\mathbf{280333.48 \pm 69704.31}$ |
| zaxxon | $56059.14 \pm 3217.77$ | $66099.28 \pm 8520.19$ | $\mathbf{67611.78 \pm 6226.04}$ |

Table 7: Final scores per game in our ablation study after 5B frames. We consider versions of CPT where the pre-trained policy is used for exploitation, exploration, or both.

| Game | $\epsilon$-greedy explore | $\epsilon z$-greedy explore | CPT (exploration only) | CPT (exploitation only) | CPT |
|---|---|---|---|---|---|
| asterix | $347585.48 \pm 244661.28$ | $414183.29 \pm 160442.25$ | $\mathbf{717259.64 \pm 94011.25}$ | $577034.99 \pm 90735.79$ | $649740.02 \pm 56368.04$ |
| bank heist | $987.94 \pm 163.14$ | $1050.67 \pm 143.36$ | $\mathbf{13881.28 \pm 2693.54}$ | $11417.81 \pm 7006.75$ | $11894.55 \pm 984.46$ |
| frostbite | $9312.19 \pm 286.04$ | $8228.56 \pm 1182.64$ | $9132.02 \pm 372.76$ | $10650.34 \pm 3690.80$ | $\mathbf{17895.70 \pm 2683.33}$ |
| gravitar | $6169.63 \pm 83.68$ | $5781.13 \pm 757.18$ | $6473.56 \pm 132.16$ | $7231.54 \pm 1994.41$ | $\mathbf{8114.29 \pm 1027.20}$ |
| jamesbond | $\mathbf{5771.41 \pm 2084.84}$ | $3633.35 \pm 272.19$ | $1713.43 \pm 477.47$ | $3898.66 \pm 1044.03$ | $1567.71 \pm 472.73$ |
| montezuma revenge | $1483.33 \pm 1118.65$ | $1465.77 \pm 1102.87$ | $11216.13 \pm 837.50$ | $6433.33 \pm 372.68$ | $\mathbf{13429.28 \pm 413.32}$ |
| ms pacman | $\mathbf{11406.77 \pm 121.72}$ | $8099.46 \pm 868.26$ | $10622.99 \pm 504.86$ | $10611.56 \pm 821.30$ | $10983.94 \pm 665.04$ |
| pong | $\mathbf{20.95 \pm 0.02}$ | $20.49 \pm 0.06$ | $20.88 \pm 0.03$ | $20.94 \pm 0.05$ | $20.89 \pm 0.07$ |
| private eye | $23589.23 \pm 11880.08$ | $\mathbf{50907.99 \pm 15174.57}$ | $40483.83 \pm 18.21$ | $37028.70 \pm 2449.66$ | $40437.40 \pm 47.86$ |
| space invaders | $3617.12 \pm 13.38$ | $3559.49 \pm 27.29$ | $\mathbf{31513.65 \pm 1093.84}$ | $3601.02 \pm 38.46$ | $27853.60 \pm 5839.58$ |
| tennis | $8.02 \pm 22.58$ | $7.95 \pm 22.55$ | $7.95 \pm 22.57$ | $-7.62 \pm 22.36$ | $\mathbf{23.96 \pm 0.03}$ |
| up n down | $527933.55 \pm 23035.67$ | $383415.21 \pm 79529.79$ | $562248.72 \pm 11617.29$ | $549769.30 \pm 3015.66$ | $\mathbf{568148.23 \pm 2861.37}$ |

Table 8: Final scores per task in Atari games with modified reward functions. We report training results for the standard game reward, a variant with sparse rewards (easy), and a task with deceptive rewards (hard). Despite the pre-trained policy might obtain low or even negative scores in some of the tasks, committing to its exploratory behavior eventually lets the agent discover strategies that lead to high returns.

| Game | $\epsilon$-greedy explore | $\epsilon z$-greedy explore | Fine-tuning | CPT (unsup) | CPT |
|---|---|---|---|---|---|
| Ms Pacman: original | $\mathbf{11407 \pm 122}$ | $8099 \pm 868$ | $8359 \pm 2117$ | $1360$ | $10984 \pm 665$ |
| Ms Pacman: ghosts (easy) | $8375 \pm 577$ | $4322 \pm 932$ | $8356 \pm 551$ | $146$ | $\mathbf{8789 \pm 651}$ |
| Ms Pacman: ghosts (hard) | $2836 \pm 26$ | $4018 \pm 1025$ | $1891 \pm 1342$ | $-898$ | $\mathbf{7868 \pm 1085}$ |
| Hero: original | $\mathbf{43762 \pm 4918}$ | $39018 \pm 3262$ | $46848 \pm 1199$ | $9298$ | $42675 \pm 3905$ |
| Hero: miners (easy) | $3000 \pm 0$ | $3000 \pm 0$ | $3000 \pm 0$ | $1351$ | $\mathbf{4665 \pm 470}$ |
| Hero: miners (hard) | $2677 \pm 23$ | $2155 \pm 95$ | $700 \pm 0$ | $-1473$ | $\mathbf{3547 \pm 122}$ |

# H  LEARNING CURVES

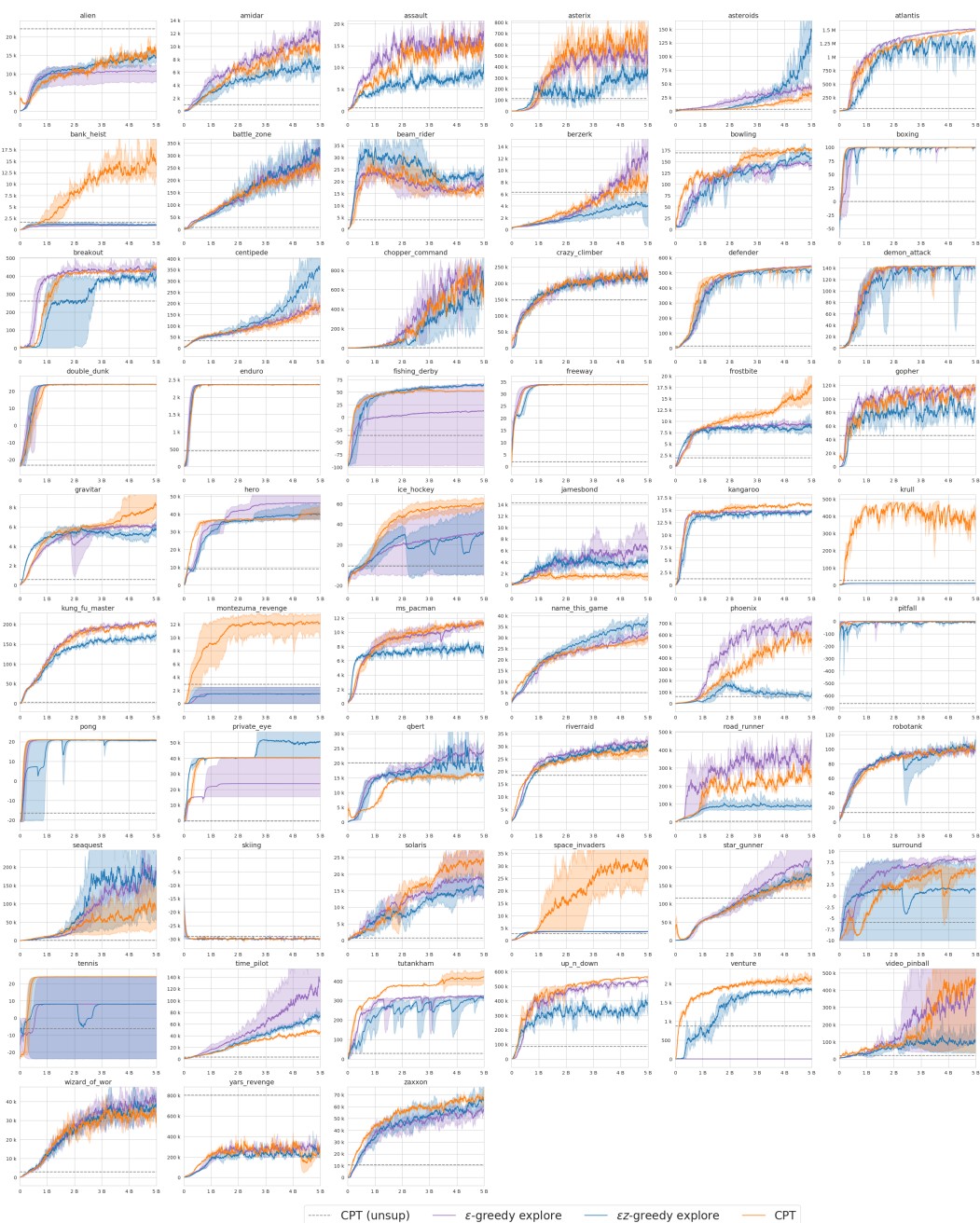

Figure 7: Training curves in all 57 Atari games after 5B frames. Shading shows maximum and minimum over 3 runs, while dark lines indicate the mean. Leveraging the pre-trained policy provides important gains, particularly in hard exploration games such as Montezuma's Revenge, while maintaining performance in games with denser rewards such as Pong or Asterix.

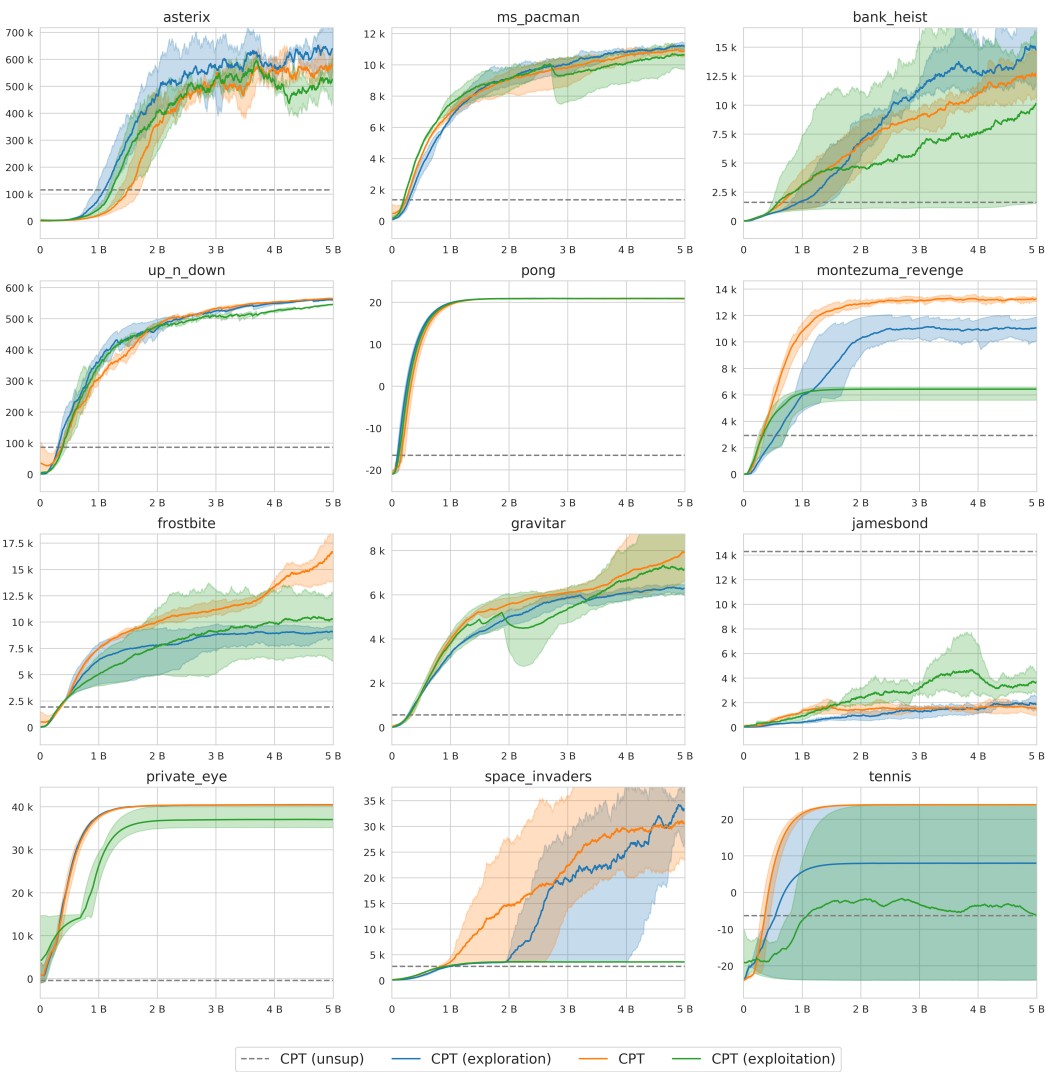

Figure 8: Training curves for ablation experiments after 5B frames. Shading shows maximum and minimum over 3 runs, while dark lines indicate the mean. Both methods offer benefits over the baselines, but in different sets of games. Combining them retains the best of both methods, and boosts performance even further in some games.

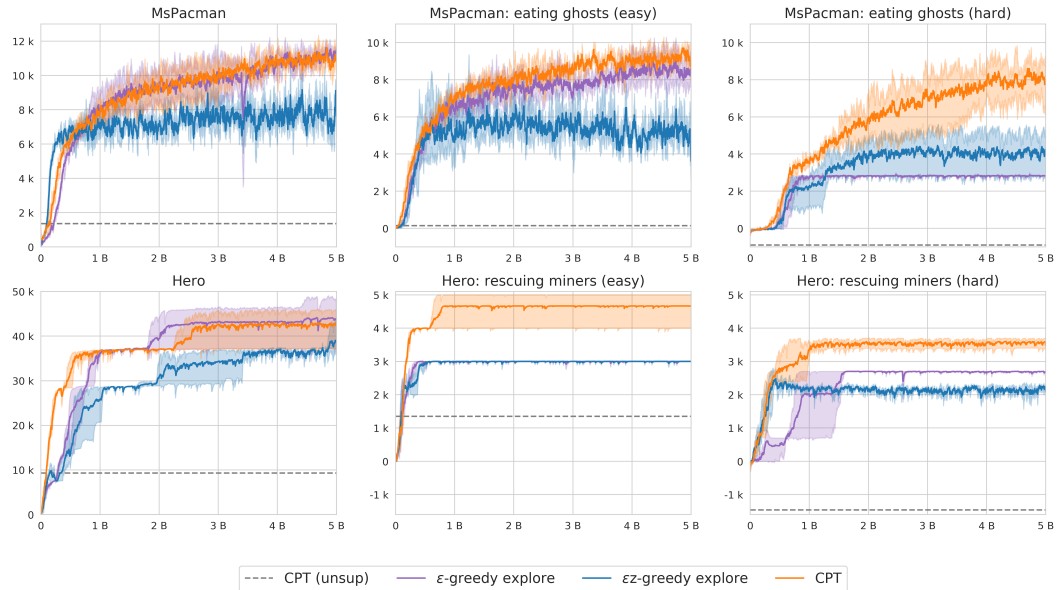

Figure 9: Alternative reward functions for MsPacman (top) and Hero (bottom). We report training curves for the standard game reward (left), a variant with sparse rewards (center), and a task with deceptive rewards (right). Despite the pre-trained policy might obtain low or even negative scores in some of the tasks, committing to its exploratory behavior eventually lets the agent discover strategies that lead to high returns.

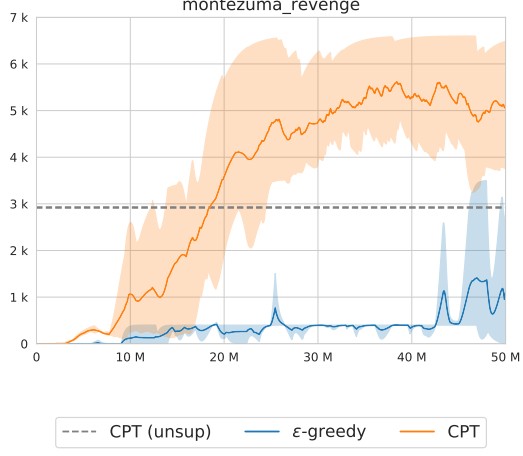

Figure 10: Training curves after 50M frames on Montezuma's Revenge, using 16 actors and the CNN encoder from the pre-trained policy. Pre-trained weights are not fine-tuned.

# I    INFLUENCE OF PRE-TRAINED POLICY

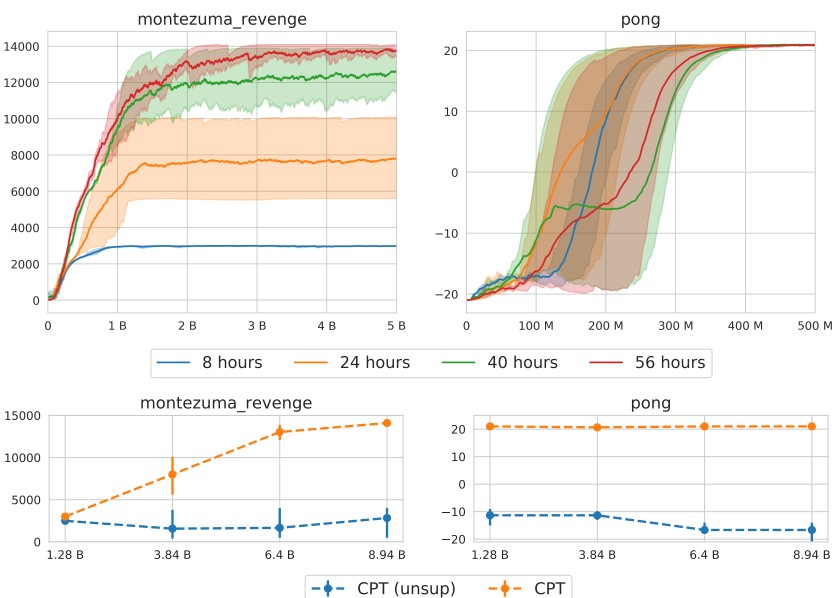

Figure 11: Influence of the pre-trained policy on the end performance. We evaluate different pre-training times: 8, 24, 40 and 56 hours of pre-training. 8 hours of training correspond roughly to 1.3B environmental steps. We show learning curves (top) and end performance (bottom) for the games of Montezuma's Revenge and Pong. When comparing end performance, we show the performance of $\pi_i$ and $\pi_e$ (after 5B environmental steps of adaptation). Longer pre-training times lead to policies that cover more of the environment. As expected, the performance in Pong is independent of the quality of the pre-trained policy while for Montezuma's Revenge, longer pre-training times lead to dramatic improvements after the adaptation stage.

## J  DISTRIBUTED SETTING

All experiments are run using a distributed setting. The evaluation we do is also identical to the one done in R2D2 (Kapturowski et al., 2019): parallel evaluation workers, which share weights with actors and learners, run the Q-network against the environment. This worker and all the actor workers are the two types of workers that draw samples from the environment. For Atari, we apply the standard DQN pre-processing, as used in R2D2. The next subsections describe how actors, evaluators, and learner are run in each stage.

### J.1  UNSUPERVISED STAGE

The computation of the intrinsic NGU reward, $r_t^{\mathrm{NGU}}$, follows the method described in Puig-domènech Badia et al. (2020b, Appendix A.1). In particular, we use the version that combines episodic intrinsic rewards with intrinsic reward from Random Network Distillation (RND) (Burda et al., 2018b).

**Learner**

- Sample from the replay buffer a sequence of intrinsic rewards $r_t^{\mathrm{NGU}}$, observations $x$ and actions $a$.
- Use Q-network to learn from $(r_t^{\mathrm{NGU}}, x, a)$ with Retrace (Munos et al., 2016) using the procedure used by R2D2.
- Use last 5 frames of the sampled sequences to train the action prediction network in NGU. This means that, for every batch of sequences, all time steps are used to train the RL loss, whereas only 5 time steps per sequence are used to optimize the action prediction loss.
- Use last 5 frames of the sampled sequences to train the predictor of RND.

**Evaluator and Actor**

- Obtain $x_t$ and $r_{t-1}^{\mathrm{NGU}}$.
- With these inputs, compute forward pass of R2D2 to obtain $a_t$.
- With $x_t$, compute $r_t^{\mathrm{NGU}}$ using the embedding network in NGU.
- (actor) Insert $x_t$, $a_t$ and $r_t^{\mathrm{NGU}}$ in the replay buffer.
- Step on the environment with $a_t$.

**Distributed training**

As in R2D2, we train the agent with a single GPU-based learner and a fixed discount factor $\gamma$. All actors collect experience using the same policy, but with a different value of $\epsilon$. This differs from the original NGU agent, where each actor runs a policy with a different degree of exploratory behavior and discount factor.

In the replay buffer, we store fixed-length sequences of $(x, a, r)$ tuples. These sequences never cross episode boundaries. Given a single batch of trajectories we unroll both online and target networks on the same sequence of states to generate value estimates. We use prioritized experience replay. We followed the same prioritization scheme proposed in Kapturowski et al. (2019) using a mixture of max and mean of the TD-errors with priority exponent $\eta = 1.0$.

### J.2  ADAPTATION STAGE IN CPT

**Learner**

- Sample from the replay buffer a sequence of extrinsic rewards $r_t$, observations $x$ and actions $a$.
- (expanded action set) Duplicate transitions collected with $\pi_p$ and relabel the duplicates with the primitive action taken by $\pi_p$ when acting.

- Use Q-network to learn from $(r_t, x, a)$ with Peng's Q($\lambda$) (Peng & Williams, 1994) using the procedure used by R2D2.

**Actor**

- (once per episode) Sample $\epsilon_{\text{levy}}$.
- Obtain $x_t$.
- If not on a flight, start one with probability $\epsilon_{\text{levy}}$.
- If on a flight, compute forward pass with $\pi_p$ to obtain $a_t$. Otherwise, compute forward pass of R2D2 to obtain $a_t$. If $a_t = |\mathcal{A}| + 1$, $a_t \leftarrow \pi_p(x)$.
- Insert $x_t$, $a_t$ and $r_t$ in the replay buffer.
- Step on the environment with $a_t$.

**Evaluator**

- Obtain $x_t$.
- Compute forward pass of R2D2 to obtain $a_t$. If $a_t = |\mathcal{A}| + 1$, $a_t \leftarrow \pi_p(x)$.
- Step on the environment with $a_t$.

**Distributed training**

As in R2D2, we train the agent with a single GPU-based learner and a fixed discount factor $\gamma$. All actors collect experience using the same policy, but with a different value of $\epsilon$.

In the replay buffer, we store fixed-length sequences of $(x, a, r)$ tuples. These sequences never cross episode boundaries. Given a single batch of trajectories we unroll both online and target networks on the same sequence of states to generate value estimates. We use prioritized experience replay. We followed the same prioritization scheme proposed in Kapturowski et al. (2019) using a mixture of max and mean of the TD-errors with priority exponent $\eta = 1.0$.

