# OpenReview forum: "Coverage as a Principle for Discovering Transferable Behavior in Reinforcement Learning"
_ICLR.cc/2021/Conference — Reject_

### Official Review · AnonReviewer3 · 2020-10-27
**Well motivated work with strong experimental results**

**Rating:** 8
**Confidence:** 3

**Review:**

This paper proposed a transfer approach for reinforcement learning. The proposed approach leverages a policy pre-trained via Never Give Up (NGU) approach, and can facilitate learning challenging RL tasks including the ones with sparse reward. This paper presents many strong pieces of evidence that this approach can be used to tackle challenging RL problems including hard exploration and multi-task learning.

Strengths
* Necessity of transferring behavior is well motivated and experiment showed its strength in challenging RL benchmarks.
* Figure 2 is useful to have intuition on how the proposed transfer method can be useful
* Table 1 set a strong unsupervised RL performance for Atari Suite
* Ablation study in Figure 3 is thought-provoking. It is interesting that the significant gain of pretraining come only when both exploitation and exploration method are used jointly.
* Figure 4 provides a useful intuition that more pretraining is beneficial for transfer to hard exploration task.

Weaknesses
* The "flights" technique is not described in detail in the main text. I managed to find the detail in Appendix A, but the pointer does not exist in the main text.
* The paper claim "coverage" as the desired objective for RL pretraining and tried to support this claim by showing the transfer performance after pretraining via Never Give Up (NGU). I am convinced that NGU is a good pre-training objective but, it is not clear whether a more general claim for "coverage" is supported as well. It is not clear whether NGU is optimizing "coverage" well, and the relation between "coverage" and transfer performance is not studied.

Comment / Questions to author
* "but little research has been conducted towards leveraging the acquired knowledge once the agent is exposed to extrinsic reward": I'm not sure whether I agree with this description. My understanding is that (Burda et al., 2018) studied a setting where an intrinsic reward is jointly used with extrinsic rewards. The only difference with this work is that the previous work did not study a setting with a clear separation between "pre-training" and "transfer".
* Is there a difference between "ez-greedy with expended action set A+ (using pre-trained policy)" vs "the proposed transfer method (exploitation + exploration)"?
* I'm curious about the comparison between CPT vs joint training with extrinsic reward. How authors would compare CPT vs joint training?

Recommendation
I recommend accepting this paper because this paper presented strong evidence that unsupervised pre-training and transfer may be a powerful approach to solve many challenging RL problems. I believe this observation is likely to catalyze future research of the related approaches, and the proposed method itself may be used for different domains to improve the capability of RL in general.

---

> ### Author Response · Authors · 2020-11-20
> **Response to AnonReviewer3**
>
> We are grateful for the feedback. Please find detailed comments below.
>
> **Details of the “flights” technique:** thanks for pointing out that we missed including a pointer to the pseudo-code in Appendix A in the main text. We have extended the description of the flights in the main text, and added the pointer. The updated version of the manuscript also includes pseudo-code for the full CPT and the ablated version that only uses the augmented action set. We included pointers to all of these in the main text.
>
> **Difference with Burda et al. (2018):** Burda et al. (2018) study a purely unsupervised setting (i.e. as we do in Section 5.1). The intrinsic rewards considered in their work had been previously used jointly with extrinsic rewards. While the difference with the two stage pre-training+transfer) setting that we use in this work might be subtle at first. This setup allows to leverage the knowledge acquired during task-agnostic pre-training stage to solve multiple tasks, as we show in our experiments (c.f. Figure 5). When optimizing extrinsic and intrinsic rewards jointly, the exploratory behavior induced by the intrinsic reward would need to be rediscovered for every downstream task.
>
> **Difference between "ez-greedy with expended action set A+ (using pre-trained policy)" vs "the proposed transfer method (exploitation + exploration)":** CPT uses the pre-trained policy to drive exploration during the flights as well as an extra action. In our ablation experiments, “CPT (exploitation only)” uses the expanded action set but relies on e-greedy for exploration. In the latter, one could replace e-greedy with ez-greedy, which explores by repeating the same action during the whole duration of the flight. The exploratory flights in CPT can be understood as a version of ez-greedy that uses a state-dependent action, $\pi_p(s)$, as the exploration option. Please let us know if this is still unclear.
>
> **CPT vs joint training with extrinsic reward:** as mentioned above, one of the advantages of pre-training+transfer over joint training is its task-agnostic aspect, which lets us reuse pre-trained policies for multiple downstream tasks. There is another subtle difference that has to do with the alignment of reward functions: joint training optimizes $r_{ext} + \beta*r_{int}$, whereas CPT optimizes $r_e$. If there is an important misalignment between $r_{ext}$ and $r_{int}$, joint training will naturally lead to a compromised solution which might be far from the optimal policy for the task. On the other hand, CPT can be more robust to such misalignment as it can ignore the pre-trained policy. This can be seen in games like Pong and Boxing (c.f. learning curves in Appendix H), where the NGU policy is bad at the downstream task and CPT can match the baselines by simply ignoring the pre-trained policy.

---

### Official Review · AnonReviewer1 · 2020-10-29
**The paper does not provide clear description of the method, despite encouraging results**

**Rating:** 5
**Confidence:** 4

**Review:**

This work proposes a methodology to use a pre-training phase (latent learning) where the agent tries to maximize the coverage of the state space to then bootstrap task solving where the agent focuses on the task-defined reward.

The paper doesn’t match conference standards in terms of description of the method. There is no clear explanation (algorithm ? figure?) of the process, how the pre-trained phase and the next phase interact, nor which information is transferred (behaviour = policy ?). No proper definition of coverage is provided. Is the pre-training phase a time when your agent learns based on a different (intrinsic) reward function ? If yes, what is this function? Is it only NGU ?
If I understood correctly, the contribution of the work is limited to training NGU without external reward, then using this pretraining to initialize the agent in an exploitation phase.

One critical question that is not answered: if you gather data in a pre-training phase, why don’t you estimate a (transition) model ? The properties of model-based RL are quite interesting in terms of finding a more optimal policy compared to learning a reactive policy. They might be arguments in favour of keeping a model-free approach but they are not presented.

The results are in accordance with the claim of the authors that their approach, favouring exploration, improves the final performance. However, you do not evaluate the actual coverage of the agent. Reward itself is a coarse metric, is suitable for the agent performance, but it is a weak claim to say that the differences in performance are due to better coverage. Please provide a measure of the improved coverage to support your claim.

In conclusion, the paper lacks a proper structure to explain the method, make it reproducible by readers; it is unclear what the contribution is, and it seems quite limited. This flaw makes it difficult to understand the results (that are nevertheless supporting the claims).

---

> ### Author Response · Authors · 2020-11-20
> **Response to AnonReviewer1**
>
> We are grateful for the feedback. We addressed the reviewers’ concerns in the updated version of the manuscript. Please find detailed comments below.
>
> **Description of the method (clarity):** CPT is described in the second paragraph of Section 2, with Figure 2 providing intuition for why this type of approach could be beneficial. Since some concerns were raised about the clarity in this description, we have updated the manuscript to improve this aspect. We added a new section before the experiments, titled “CPT: Coverage Pre-training for Transfer”, that describes the proposed method clearly. CPT does not initialize the weights of the new policy using the pre-trained one, and knowledge is transferred through behavior instead (i.e. observation->action mappings in the learned policies). We have modified the manuscript to make these points clearer in both Section 3 (Leveraging Pre-trained Policies) and the new Section 5 (CPT). The original submission included pseudo-code for the flights in Appendix A, but the main text was missing a pointer to the appendix as noted by R3. We have added this pointer to the updated manuscript, and extended the description of how the flights are generated in the main text. We have also included pseudo-code for the CPT actor that uses both the augmented action set and the exploratory flights, as well as for the ablated version that uses the augmented action set only.
>
> **CPT:** We believe there has been a misunderstanding in how CPT works, and we hope that the updated manuscript addresses this issue. We would like to summarize the important aspects of CPT here as well. First, we pre-train exploratory policies to maximise the expected discounted sum of intrinsic rewards as defined in NGU. When solving downstream tasks, we randomly initialize a new policy (i.e. there is no weight transfer from the pre-trained NGU policy) and leverage the pre-trained policy through the two mechanisms described in Section 3 (namely, using the augmented action set and exploratory flights). One of the main goals of this work is to highlight the impact of transferring behavior (not only representations or weights) in the RL problem, contrasting results in the semi-supervised setting. The submission includes some experiments where we load the CNN that was pre-trained in the unsupervised stage, namely Figure 1 (behavior+representation) and the data-efficiency experiments on Montezuma’s Revenge, to show that the gains provided by CPT are complementary to those from standard transfer strategies that use pre-trained weights. With this, we would like to stress that our contributions are not “limited to training NGU without external reward, then using this pretraining to initialize the agent in an exploitation phase”. To further prove this point, we have added a fine-tuning baseline to Figures 1 and 3. Fine-tuning the pre-trained policy clearly underperforms with respect to the proposed method.
>
> **Learning a model:** Previous literature studying the two-stage setting in our work has mostly considered model-free approaches. There are some exceptions, like the World Models work by Ha & Schmidhuber, which considers smaller scale experiments and environments. We agree that this is an interesting direction for future research, and have included a mention to model-based approaches in the review of strategies for leveraging unsupervised interaction (c.f. third paragraph in Section 1). One advantage of model-free approaches like ours is that bad policies are easier to ignore than bad models. This can be seen in games like Pong and Boxing (c.f. learning curves in Appendix H), where the NGU policy is bad at the downstream task and CPT can match the baselines by simply ignoring the pre-trained policy.

---

> > ### Author Response · Authors · 2020-11-20
> > **Response to AnonReviewer1 (part 2)**
> >
> > **Measuring coverage:** The concept of coverage that we consider in this work makes reference to the ability of an agent to visit as many states as possible within a single episode. This is slightly different to what most intrinsic curiosity methods achieve, which encourage agents to visit many different states during training. However, this does not guarantee that the resulting policy will visit that many states on a per episode basis (c.f. Lee et al. [1] for a detailed explanation). Measuring coverage is an open research problem, and some previous works have used proxy metrics for this. For instance, Hazan et al. (2019) used the number of unique (x,y) positions visited by the agent in locomotion environments. We counted the number of distinct rooms visited by the policies in Montezuma’s Revenge, where CPT visits 13 different rooms while ez-greedy exploration only discovers 2 rooms.
> >
> > The results support the claims: we appreciate that the reviewer acknowledged the strong experimental results, and we hope that the updates made to the manuscript addressed the clarity issues and made our contributions clearer.
> >
> > [1] Lee, L., Eysenbach, B., Parisotto, E., Xing, E., Levine, S., & Salakhutdinov, R. (2019). Efficient exploration via state marginal matching. arXiv preprint arXiv:1906.05274.

---

### Official Review · AnonReviewer4 · 2020-10-29

**Rating:** 4
**Confidence:** 4

**Review:**

The paper studies a pre-training approach to reinforcement learning. The objective is, first to pre-train a model considering that, without reward, interaction with an environment is cheap, and second, to fine-tune a policy given a particular reward function.

As a first contribution, the paper proposes two strategies to use a pre-trained policy for discovering an efficient task-dependent policy: i) the action set is expanded such that the policy can choose to follow the pre-trained policy and ii) exploration may be done by following the pre-trained policy on t timesteps, t being randomly chosen. As a second contribution, the paper proposes a criterion to pre-train a policy based on a coverage criterion.  The principle is to encourage a policy to generate trajectories that are going through as many states as possible.

Experiments are made on different environments. First, the pre-trained policies are evaluated based on how much reward they are able to collect, and compared to other unsupervised approaches. Second, the final policy is compared to epsilon-greedy and epsilon-z-greedy approaches. In the two cases, the proposed approach outperform the baselines.

== Comments:
First of all, the paper clearly lacks of details, and it is difficult to be sure about what is really done, and what is the final algorithms. As far as I understand, instead of proposing a really new approach, the paper is more stacking two approaches (i.e NGU and R2D2, just changing a little bit the action space) and it does not really provide any justification about what is done. From my point of view, it is more a paper investigating if coverage may be a good unsupervised training criterion than a paper presenting a new model.

Concerning the exploration/exploitation of pre-train policies, the paper does not really describe how the 'flights' are generated (section 3). I would advise the authors to provide more details on this aspect of their algorithm.

Concerning the coverage approach, I am not convinced that the paper allows us to draw any conclusion on the interest of using such a criterion during the pre-training phase. Indeed, the authors are mainly evaluating their approach on Atari games, on which there is a clear relationship between the length of the episodes and the final score achieved by an agent. This is what is shown in Section 5.1: coverage is a good surrogate objective for solving atari games.  The article is thus lacking evaluation on other types of environments, and the performance obtained by the model on Atari games is mainly due to the use of the NGU model which has been developed more specifically for this type of environment.

To conclude, I think that the paper is failing to provide evidence that the coverage approach is a good approach to unsupervised pre-training of policies. Moreover, I have the feeling that the coverage criterion may be good for particular types of environments (like Atari games), but not for some others, making the proposed approach very specific to particular problems. Combined with the lack of novelty, and the lack of details, I recommend to reject the paper.

Considering the answers from the authors, I decided to not change my score.

---

> ### Author Response · Authors · 2020-11-20
> **Response to AnonReviewer4**
>
> We are grateful for the feedback. We addressed the reviewers’ concerns in the updated version of the manuscript. Please find detailed comments below.
>
> **"instead of proposing a really new approach, the paper is more stacking two approaches (i.e NGU and R2D2, just changing a little bit the action space)" / "From my point of view, it is more a paper investigating if coverage may be a good unsupervised training criterion than a paper presenting a new model".** The paper addresses the problem of reinforcement learning with task-agnostic pre-training as presented by Hansen et al 2020 (reference from the manuscript). Neither R2D2 nor NGU (which is an extension of R2D2) are methods designed for this task. This setting presents two main challenges: (1) defining a pretext task for acquiring knowledge in the absence of task-specific rewards, and (2) efficiently using that knowledge when the reward signal is exposed. As noted by the reviewer, one of the contributions of this submission is to explore the potential of coverage as the training criterion for (1). We further propose to use NGU for this task.
>
> We want to highlight that even when using NGU for solving (1), problem (2) is by no means straightforward.  NGU proposes to jointly train a family of policies with different degrees of exploratory behaviour. NGU treats the training of exploratory policies as auxiliary tasks that help to build a shared architecture for the family. Transfer is by design obtained via joint training. Furthermore, an important difference with the current work is that in NGU the exploratory policies are task-dependent: they are trained to maximise a different mixture of extrinsic and intrinsic returns. Rather than adapting NGU to the transfer domain, we propose a generic way of leveraging pre-trained policies that explicitly accounts for the transfer of behavior, a feature that (to the best of our knowledge) has not received significant attention in the literature. The reviewer mentions that the proposed second stage is “to fine-tune a policy given a particular reward function”. We want to highlight that this is not the case, please see below.
>
> The main point of this work is to highlight that transfer in the RL setting can benefit greatly by explicitly reusing pre-trained knowledge in the form of behavior. To deliver this point cleanly, we propose a strategy to leverage behavior (i.e. mapping from observations to actions) rather than pre-trained neural network weights. All CPT experiments use randomly initialized networks, except for those where we study whether reusing representations (in the form of the CNN encoder) provide efficiency boosts. We conclude that the transfer of weights and behavior are complementary. The details of the architecture in the adaptation stage is depicted in Appendix F. A particular way of transfering weights is by using a simple fine-tuning mechanism, as mentioned by the reviewer. Our point is that fine-tuning is not enough in the RL setting. The main reason is that the process of fine-tuning might “destroy” the knowledge acquired in the pre-training stage (a form of catastrophic forgetting), before it is fully transferred.  We acknowledge that the question by the reviewer is relevant and experimental evaluation of this statement is required. We included a new set of experiments where the pre-trained NGU policy is fine-tuned when exposed to rewards. This new fine-tuning baseline has been added to Figures 1 and 3. Fine-tuning the pre-trained policy clearly underperforms with respect to the proposed method, which we believe demonstrates the need for alternative transfer strategies like the ones proposed in this submission. Furthermore, we could incorporate fine-tuning in CPT in a natural way: training via CPT a network initialised to be equal to the pre-trained one (instead of random weights). This was not straightforward to implement in our current codebase, which is why we did not report results in the rebuttal, but we are working to have these ready for the camera-ready version of the paper.

---

> > ### Author Response · Authors · 2020-11-20
> > **Response to AnonReviewer4 (part 2)**
> >
> > The proposed adaptation method uses the pre-trained policy for both exploitation (through an augmented action set) and exploration (through exploratory flights). CPT consists in pre-training for coverage and then leveraging the resulting policy for both exploitation and exploration when a task reward is exposed. The submission included several ablation studies to provide insight on the contribution of each technique, but the proposed method combines both strategies. We employ the intrinsic NGU reward for the purpose of maximizing coverage in the pre-training stage due to its scalability to environments that involve e.g. pixel observations or partial observability. However, we could employ any alternative method that yields exploratory policies that visit as many states as possible.
> >
> > Having said this, we acknowledge that our transfer approach does involve relatively simple changes such as augmenting the action space and changing the exploration policy. But discovering which changes are needed for successful transfer was highly non-trivial and the resulting simplicity should be seen as a strength: it significantly outperforms the baselines while being easy to implement.
> >
> > We use the distributed R2D2 agent to optimize our policies in both pre-training and adaptation stages. This choice allows for faster iteration, but the proposed strategies are independent of the underlying RL agent as long as it can learn from off-policy data. For this reason, we disagree with the statement that “the paper is more stacking two approaches (i.e NGU and R2D2, just changing a little bit the action space)”.
> >
> > **"the paper clearly lacks of details, and it is difficult to be sure about what is really done, and what is the final algorithms" / "I would advise the authors to provide more details on this aspect of their algorithm".** The original submission included pseudo-code for the flights in Appendix A, but the main text was missing a pointer to the appendix as noted by R3. We have added this pointer to the updated manuscript, and extended the description of how the flights are generated in the main text. We have also included pseudo-code for the CPT actor that uses both the augmented action set and the exploratory flights, as well as for the ablated version that uses the augmented action set only.
> >
> > The paper makes two claims regarding coverage: (1) it is a good criterion for exploration, and (2) it might be a good strategy for exploitation in some cases. Coverage has been used as a criterion for exploration in theoretical works (e.g. Kearns & Singh (2002) and Liu & Brunskill (2018), in the manuscript) and more practical approaches (e.g. eigenoptions [1], covering options [2]). The fact that coverage sometimes correlates with performance in many Atari games had also been shown by Burda et al. (2018).
> >
> > **"the performance obtained by the model on Atari games is mainly due to the use of the NGU model which has been developed more specifically for this type of environment".** While unsupervised NGU achieves very high scores in the Atari benchmark, we would like to highlight some of the evidence in the paper showing that CPT can achieve high performance even when the pre-trained policy via NGU (this is, covering the space) is not well aligned with the task reward. Even in the cases in which the NGU policy is bad for the task, we do achieve good results thanks to the fact that the NGU policy is able to explore well (cover the space). After adaptation, CPT achieves very high scores in games where the unsupervised policy does not perform well, such as Boxing or Pong (c.f. learning curves in Appendix H). With the goal of removing the correlation between exploration and game scores, we designed alternative reward functions that penalize exploratory behaviors. Even in such adversarial situation, where the NGU policy achieves low or even negative scores, CPT outperforms the baselines by an important margin (c.f. Figure 5).  A policy maximising coverage is good to help the agent “find” rewards, while the policy itself might not be very performant on the task.
> >
> >
> > [1] Machado, M. C., Bellemare, M. G., & Bowling, M. (2017). A laplacian framework for option discovery in reinforcement learning. ICML.
> >
> > [2] Jinnai, Y., Park, J. W., Abel, D., & Konidaris, G. (2019). Discovering options for exploration by minimizing cover time. ICML.

---

### Official Review · AnonReviewer2 · 2020-10-29
**Recommendation to Reject**

**Rating:** 4
**Confidence:** 4

**Review:**


#### Summary:

In this work the author focus on transfer in RL via proposing to transfer knowledge through behavior instead of representations. They propose using coverage as an objective for the pre-training procedure. They then employ the NGU (Badia 2020) They also propose a method based on coverage pre-training for transfer and provide empirical evidence in support of their method for transfer in RL on the Atari suite.


#### Strengths:

- The paper has a good experimentation section, including empirical analysis such as ablation studies and effect or pre-training that is insightful.

#### Weakness:

- **Novelty**: The notion of coverage has been explored in the past, for instance in [1, 2, 3]. The authors fail to differentiate what do they do in their method that fixes the limitations of the previous work.

- **Incremental Nature of work**: The proposed method relies heavily on the Never Give Up (Badia et al) for finding the policies that maximize the coverage. It is not clear what this work is proposing other than using NGU in the transfer setting directly.

- **Cost of pretraining**: The pre-training comes at a cost. In their setup, the agents are allowed as many interactions with the environment as needed. This can be extremely futile for high-dimensional state and action spaces. If the dynamics model doesn't change for the downstream tasks, then this choice of pre-training is a valid choice (that has already been explored in the previous works as the authors mention). However, if there is a change in dynamics then there can be scenarios where the pre-training procedure doesn't result in any benefits. The procedure mentioned in Section 3 is already being employed in many different previous works, and it is not clear what is the setting being considered in this work. Also in Section 3, $\pi_i$ is undefined.

- **Missing references**: The authors make statements and then fail to give the appropriate references for the same. For instance, the authors quote that "RL techniques have not yet seen the advent of a two-stage setting where task-agnostic pre-training is followed by efficient transfer to downstream tasks", however, there are multiple works that are based on essentially this approach ([4, 5]).

- **Reproducibility**: There is no mention of code release and makes me skeptical of their results, especially when this an empirical results-driven work.


#### References:

- [1] Schmidhuber, Jürgen. "Formal theory of creativity, fun, and intrinsic motivation (1990–2010)." IEEE Transactions on Autonomous Mental Development 2.3 (2010): 230-247.
- [2]  Hazan, Elad, et al. "Provably efficient maximum entropy exploration." International Conference on Machine Learning. 2019.
- [3] Conti, Edoardo, et al. "Improving exploration in evolution strategies for deep reinforcement learning via a population of novelty-seeking agents." Advances in neural information processing systems. 2018.
- [4] Lesort, Timothée, et al. "State representation learning for control: An overview." Neural Networks 108 (2018): 379-392.
- [5] Ha, D. and Schmidhuber, J. (2018). World Models. ArXiv e-prints.

---

> ### Author Response · Authors · 2020-11-20
> **Response to AnonReviewer2**
>
> We are grateful for the feedback. We addressed the reviewers’ concerns in the updated version of the manuscript. Please find detailed comments below.
>
> **Novelty**: We agree that the notion of coverage has been explored before. What we propose in our submission is using coverage to learn transferable policies, and this proposal is not tied to a particular strategy for maximizing coverage. We employ NGU for this purpose due to its scalability to environments that involve e.g. pixel observations or partial observability. Please note that [1,2,3] are not straightforward to apply to this type of setting: [1] does not define a practical algorithm that can be run on current benchmarks. [2] requires an estimate of state distributions, which is challenging for domains like Atari, thus it only considers fully observable MDPs with relatively small state dimensionality. [3] suffers from similar issues, and authors relied on privileged information (the RAM state of the Atari machine) to define the behavior characterization required by their QD methods when deployed in Atari. We have extended the last paragraph of Section 4 in order to stress the motivation behind this choice and explicitly state the difference with respect to prior work as requested by the reviewer.
>
> **Incremental nature of work**: Our implementation makes use of NGU to discover exploratory behavior to maximise coverage over a single episode. We would like to stress that the proposed method is generic and can be combined with other pre-training mechanisms. Indeed, one of the main contributions of this submission is the proposed strategy for leveraging pre-trained policies, also referred to as ‘behaviors’ in the manuscript. The proposed method treats pre-trained policies as black boxes, and only requires sampling actions from such policies (i.e. we do not use any pre-trained weights except for the experiments where we also transfer representations).
>
> Having said that, we acknowledge that the current implementation heavily relies on NGU. We would like to stress that “using NGU in the transfer setting directly”, as done in our experiments, is by no means straightforward. NGU proposes to jointly train a family of policies with different degrees of exploratory behaviour. NGU treats the training of exploratory policies as auxiliary tasks that help to build a shared architecture for the family. Transfer is by design obtained via joint training. Furthermore, an important difference with the current work is that in NGU the exploratory policies are task-dependent: they are trained to maximise a different mixture of extrinsic and intrinsic returns. Rather than adapting NGU to the transfer domain, we propose a generic way of leveraging pre-trained policies that explicitly accounts for the transfer of behavior, We show that explicit transfer of behaviour is more effective than fine-tuning the neural network weights, but it is ignored by the sole transfer of representations. We have clarified these points in the updated manuscript.
>
> **Cost of pretraining**: our method would indeed fail when the state transition dynamics change. However, it is often the case that the reward dynamics aren’t given e.g. when rewards come from expensive human judgements or the downstream task has not yet been determined. In many of these cases the state transition dynamics are stationary despite these issues. Indeed, this insight is the basis of much work, such as the successor features framework (Barreto et al. 2017, in the manuscript). The reviewer is correct, this setting assumes that unstructured interaction with the environment comes at significantly lower cost relative to interactions in the structured reward-driven adaptation stage. Note that a single pre-trained policy is expected to be used for transfer to multiple such tasks, amortizing the cost incurred during the unsupervised learning stage (e.g. in the experiments reported in Figure 5). It is worth reiterating that the unsupervised pre-training regime for Atari was created independently from this paper (Hansen et al. 2020, in the manuscript), and was established with these same assumptions in mind. We will update the paper to make the rational (and limitations) more explicit.

---

> > ### Author Response · Authors · 2020-11-20
> > **Response to AnonReviewer2 (part 2)**
> >
> > **Missing references**: We have added the suggested references. We acknowledge that model-based methods have used this type of two-stage setting in the past, and have updated the manuscript to include this fact. While those methods are promising, scaling them up to complex tasks is not trivial (e.g. the world model in [5] is unlikely to perform well on transitions that were not observed when collecting data with the random policy). Regarding the claim that "RL techniques have not yet seen the advent of a two-stage setting where task-agnostic pre-training is followed by efficient transfer to downstream tasks": we did not mean that the two-stage setting had not been studied before, but we wanted to highlight that it has not brought to RL the same benefits that have been observed in supervised settings (e.g. this setting has revolutionized fields like NLP thanks to BERT or GPT-2/3).
> >
> > **Reproducibility**: The original submission included pseudo-code for the flights in Appendix A, but the main text was missing a pointer to the appendix as noted by R3. We have added this pointer to the updated manuscript, and extended the description of how the flights are generated in the main text. We have also included pseudo-code for the CPT actor that uses both the augmented action set and the exploratory flights, as well as for the ablated version that uses the augmented action set only. Releasing code for this type of distributed settings is very challenging because it is tied to the infrastructure being used. Please let us know if anything is still unclear and we will be happy to update the submission with a clarification.

---

### Author Response · Authors · 2020-11-20
**To all reviewers: changes to the manuscript**

We thank all reviewers for their valuable suggestions and comments. We have made the following changes to our revised manuscript:

- Added results of CPT@0 to Table 2 to show adaptation gains
- Added results of CPT@0 to Figure 3 to show adaptation gains
- Replaced $\pi_i$ with $\pi_p$ in Section 3
- Added a fine-tuning baseline to Figure 1, to the ablation experiments (Figure 3), and to the experiments on games with modified reward functions (Figure 5)
- Included a new section before the experiments with a detailed description of the proposed method
- Expanded the description of the flights in Section 3
- Added pseudo-code for the actor logic in CPT and its ablated versions
- New Appendix J with details of the distributed setting in both stages
- Added clarifications and additional references throughout the manuscript

All changes are marked in red in the revised manuscript.

---

### Decision · Program_Chairs · 2021-01-07
**Final Decision**

**Decision:**

Reject

**Comment:**

The paper studies the unsupervised RL problem, where the agent is allowed to interact with the environment for a certain amount of time without any extrinsic reward. The main idea is that the initial unsupervised training phase can be used to learn a set of "skills" that could help both in exploration and zero-shot transfer for any downstream task.

There is general consensus among the reviewers that the paper is studying an important problem and that the empirical validation is solid. Nonetheless, the technical contribution and the positioning wrt to the relevant literature are relatively weak for proposing acceptance for the paper. Before entering into details, I would like to acknowledge the fact that the rebuttal and the revised version did improve the original submission and clarified some aspects (eg, the structure of the algorithm), yet the contribution does not seems strong enough.

The idea of state space coverage is indeed not novel, either for exploration or for transfer (as properly reviewed in the paper). The authors identified some weaknesses of existing methods (eg, estimating state distributions), but it remains unclear whether the algorithm they propose has any significant technical contribution. In fact, as confirmed by the authors, CPT is rather applying any algorithm that *could* perform a good state space coverage and learn policies at the same time and then use the learned policy during the downstream task combined with a relatively basic "option-level" eps-greedy strategy. In this sense, it seems like CPT overcomes the limitations of previous algorithms, just by applying another existing algorithm (NGU) that is more scalable. While the evidence that this is "enough" to obtain good results is indeed interesting, it doesn't seems like it is pushing the algorithmic state of the art forward.

A more substantial contribution would be to dive deeper into the state coverage problem and provide an algorithm that is more specifically designed for the transfer setting considered in the paper. In fact, there is no clear evidence that NGU is the *right* approach to perform good coverage and return "useful" skills. Since this is the core concern of the paper, the technical contribution should be more significant on this part. The "meta" algorithm in itself seems rather standard otherwise.